# The M$_S$ 6.9, 1980 Irpinia Earthquake from the Basement to the Surface: A Review of Tectonic Geomorphology and Geophysical Constraints, and New Data on Postseismic Deformation

**Alessandra Ascione** [1], **Sergio Nardò** [1] **and Stefano Mazzoli** [2],*

[1]   Department of Earth Sciences, Environment and Georesources (DiSTAR), University of Naples Federico II, 80126 Naples, Italy; alessandra.ascione@unina.it (A.A.); sergio.nardo@unina.it (S.N.)

[2]   School of Sciences and Technology, Geology Division, University of Camerino, 62032 Camerino, Italy

*    Correspondence: stefano.mazzoli@unicam.it

**Abstract:** The M$_S$ 6.9, 1980 Irpinia earthquake occurred in the southern Apennines, a fold and thrust belt that has been undergoing post-orogenic extension since ca. 400 kyr. The strongly anisotropic structure of fold and thrust belts like the Apennines, including late-orogenic low-angle normal faults and inherited Mesozoic extensional features besides gently dipping thrusts, result in a complex, overall layered architecture of the orogenic edifice. Effective decoupling between deep and shallow structural levels of this mountain belt is related to the strong rheological contrast produced by a fluid-saturated, shale-dominated mélange zone interposed between buried autochthonous carbonates—continuous with those exposed in the foreland to the east—and the allochthonous units. The presence of fluid reservoirs below the mélange zone is shown by a high V$_P$/V$_S$ ratio—which is a proxy for densely fractured fluid-saturated crustal volumes—recorded by seismic tomography within the buried autochthonous carbonates and the top part of the underlying basement. These crustal volumes, in which background seismicity is remarkably concentrated, are fed by fluids migrating along the major active faults. High pore fluid pressures, decreasing the yield stress, are recorded by low stress-drop values associated with the earthquakes. On the other hand, the mountain belt is characterized by substantial gas flow to the surface, recorded as both distributed soil gas emissions and vigorous gas vents. The accumulation of $CO_2$-brine within a reservoir located at hypocentral depths beneath the Irpinia region is not only interpreted to control a multiyear cyclic behavior of microseismicity, but could also play a role in ground motions detected by space-based geodetic measurements in the postseismic period. The analysis carried out in this study of persistent scatterer interferometry synthetic aperture radar (PS-InSAR) data, covering a timespan ranging from 12 to 30 years after the 1980 mainshock, points out that ground deformation has affected the Irpinia earthquake epicentral area in the last decades. These ground motions could be a result of postseismic afterslip, which is well known to occur over years or even decades after a large mainshock such as the 23 November 1980, M$_S$ 6.9 earthquake due to cycles of $CO_2$-brine accumulation at depth and its subsequent release by M$_w$ ≥ 3.5 earthquakes, or most likely by a combination of both. Postseismic afterslip controls geomorphology, topography, and surface deformation in seismically active areas such as that of the present study, characterized by ~M 7 earthquakes. Yet, this process has been largely overlooked in the case of the 1980 Irpinia earthquake, and one of the main aims of this study is to fill such the substantial gap of knowledge for the epicentral area of some of the most destructive earthquakes that have ever occurred in Italy.

**Keywords:** 1980 Irpinia earthquake; active faults; postseismic afterslip; southern Apennines

## 1. Introduction

The Irpinia M$_S$ 6.9 earthquake occurred on 23 November 1980 [1], producing vast damage and causing about 3000 fatalities. It nucleated on an approximately 60 km long, NW-SE-striking normal fault system with at least three main rupture episodes at 0 s, 20 s, and 40 s (Figure 1).

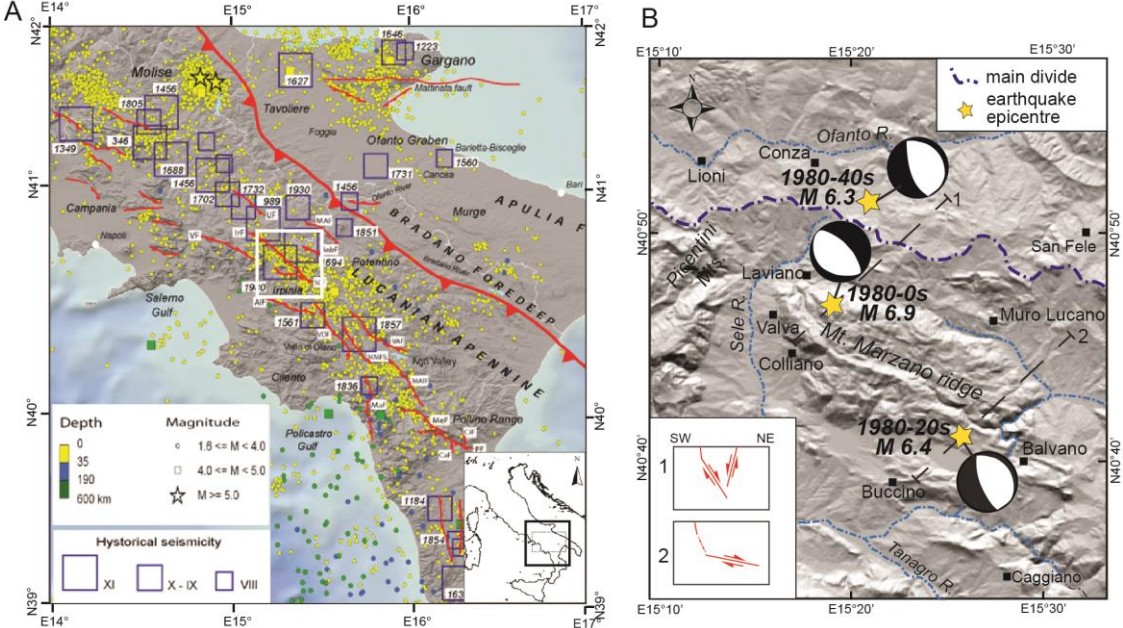

**Figure 1.** (**A**): Historical and instrumental seismicity of southern Italy (modified after Frepoli et al. [2]). The white frame indicates the location of the map in diagram B. (**B**): Epicentral locations for the three main rupture episodes at 0 s, 20 s, and 40 s, with Mw 6.9, 6.4, and 6.3, of the 1980 Irpinia earthquake; focal mechanisms from Westaway and Jackson [3]. Inset in the lower left corner are schematic sections (location in the map), indicating the fault system activated with the 1980 earthquake (redrawn from Barnard and Zollo [1]).

A crustal extensional stress regime controls earthquake generation processes in the Apennine mountain chain. Background seismicity is characterized, during the postseismic to (possibly) inter-seismic period, by micro-earthquakes (M$_L$ < 3.5) approximately confined within the same volume wherein the faults that caused the 1980 earthquake and its aftershocks were located [4–6]. The surface faults exposed in this area are decoupled from the seismically active deep-seated structures, which, in turn, reactivate inherited basement faults [7]. Effective decoupling between deep and shallow structural levels is related to the strong rheological contrast produced by the fluid-saturated, clay-rich mélange zone interposed between the foreland Apulia Platform carbonates and the allochthonous units, which include carbonate platform (i.e., Apennine Platform) and basin (i.e., Internal and Lagonegro) successions (Figure 2).

The identification of active faults and, more in general, the reconstruction of the active tectonic setting on a regional scale are crucial to the assessment and mitigation of seismic hazard and related phenomena (e.g., ground shaking, surface ruptures, landsliding, etc.), and of hazard resulting from surface deformation (e.g., flooding, subsidence, etc.). The definition of the active tectonic setting of an area requires the identification and geometric characterization of potentially active structures and the estimation of rates of activity of the identified faults. An important contribution to such definition is provided by the combination of data deriving from instrumental seismicity, from the historical record of seismicity, and from paleoseismological techniques. Despite providing information crucial to the seismic hazard prevention, such a combined approach is, however, partial due to the relatively narrow time window (in general, ≤10$^3$ y) it explores. In fact, taking into account that rates of deformation may be either slow (e.g., a long quiescence may separate large earthquakes) and/or uneven both spatially (i.e., along strike of single structures over short timespans) and temporally (on single structures over

larger timespans), short-term records may fail sampling active deformation. This may result in both an underestimation of active fault segments and an incomplete outline of the surface deformation scenario.

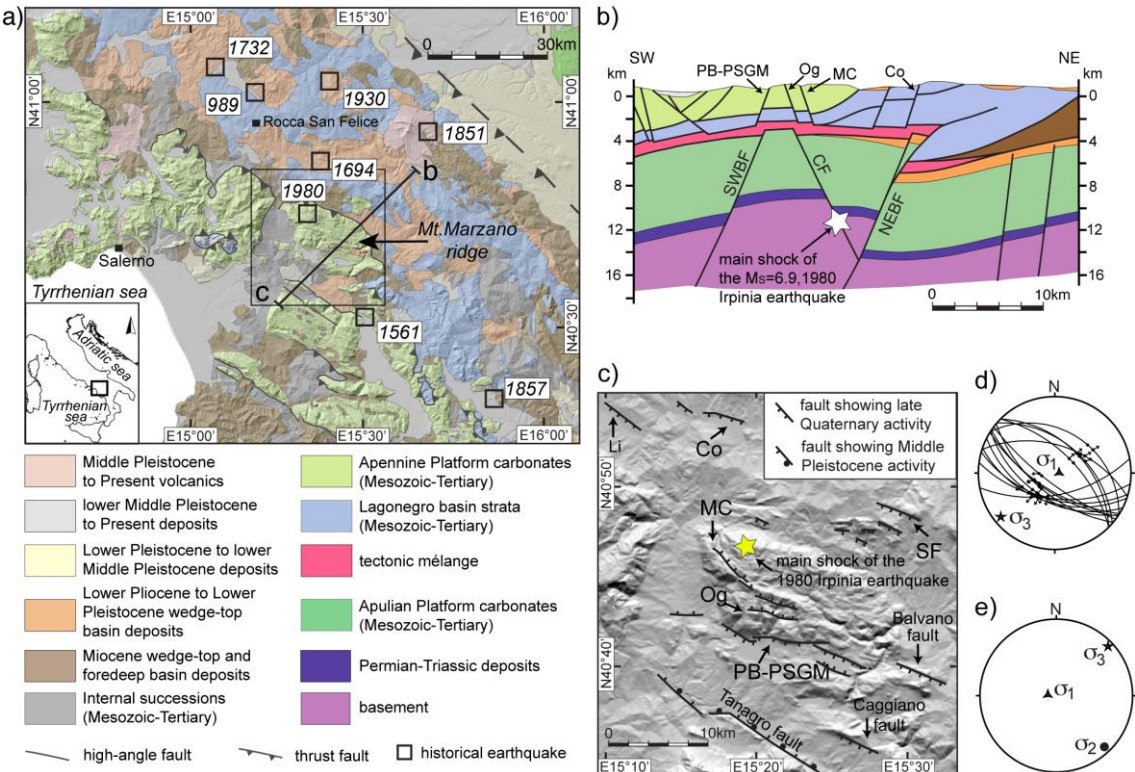

**Figure 2.** Tectonic setting of the 1980 Irpinia earthquake. (**a**) Geological sketch map of the southern Apennines, showing location of seismic stations and main historical and instrumental earthquakes. (**b**) Cross-section (after Ascione et al. [7]). Star shows hypocenter of the $M_s$ 6.9, 1980 Irpinia earthquake; SWBF: SW Boundary Fault; CF: Central Fault; NEBF: NE Boundary Fault. (**c**) Outcropping Quaternary fault arrays (modified after Ascione et al. [7]). PB-PSGM: Piani di Buccino-Pantano di San Gregorio Magno fault zone; Og: Mount Ogna fault zone; MC: Mount Marzano–Mount Carpineta fault zone; Li: Lioni fault; Co: Conza fault; SF: San Fele fault. (**d**) Results of cumulative stress inversion from outcropping active fault segments (from Ascione et al. [7]). (**e**) Results of cumulative stress inversion from earthquake focal mechanism, data are from De Matteis et al. [4] (modified from Amoroso et al. [8]).

A more comprehensive definition of the deformation scenario is obtained if information on geometry and kinematics of deep-seated active fault segments, on spatial distribution and intensity of seismic shaking, and on the behavior (in terms of recurrence and magnitude of surface offset) of single fault segments is paralleled with observations encompassing larger timescales (e.g., the last 300–400 ky, i.e., 'intermediate timescales' according to Burbank and Anderson [9]). The latter, although less detailed on the history of single structures, may provide significant insights on the spatial distribution, pattern, and average rates of surface deformation. In fact, deformation accumulates through time, leaving an increasing imprint in the landscape which may be detected by an analysis of topography and may be temporally constrained by the stratigraphic record. Although it has become common to relate the motions during earthquakes to the Quaternary geological structures and topography in the epicentral regions, the deformation preserved in structural and topographic features represents the combined effects of entire earthquake cycles rather than only the coseismic period during which earthquake slip occurs. Therefore, in order to understand the formation and evolution of geology and topography over successive earthquake cycles, it is necessary to make observations that cover as wide a range of the cycle as possible.

The 23 November 1980, $M_S$ = 6.9 Irpinia earthquake, the strongest and most destructive ($I_0$ = X MCS) seismic event of the last decennia in southern Italy, affected the Mt. Marzano area with widespread coseismic ruptures (e.g., [1,10–17]). The 1980 Irpinia earthquake strongly impacted the state of knowledge on seismicity of Italy, primarily because it was the first Italian earthquake for which coseismic surface faulting was assessed at two sites (namely, the Piano di Pecore and Pantano di San Gregorio Magno sites) by means of just posteventum surveys [18,19]. In addition, with further studies, a 38 km long alignment of surface faulting was recognized, in agreement with the inferred length of the rupture at depth [12]. The history of surface faulting in the last millennia at Piano di Pecore was assessed with the first paleoseismological study in the Italian territory [13].

In the last 40 years, many studies have provided constraints to seismicity and tectonics active in the 1980 event epicentral areas. According to such studies, it appears that both the earthquake complexity and the historical and present-day seismicity represent the response to a complex active tectonics scenario and suggest the existence of an 'active' graben-like structure defined by at least two antithetic major faults, which were originally identified by Barnard and Zollo [1] (Figure 1). However, the pattern, distribution, and localization of active structures, is still debated and, in some instances, controversial, with major implications on both cumulative deformation rates and related hazard. In fact, the active tectonic framework of the Irpinia earthquake epicenter area has been related to one or more structures substantially following the surface ruptures originally identified by Pantosti and Valensise [12,20–22], while a more articulated pattern composed by several normal faults arrays showing substantially coeval late Quaternary activity and spanning over the seismologically identified graben structure has been identified by detailed geomorphological analyses integrated by age constraints on the Quaternary deposits [7]. Further constraints on fault activity were obtained by Ciotoli et al. [23] using soil gas trace and carrier species as potential tracers of fault systems in the Piano di Pecore, a small-sized tectono-karstic basin located in the northern Mt. Marzano massif, which was affected by coseismic surface faulting of the 1980 Irpinia earthquake. The coseismic rupture zone in Piano di Pecore was investigated by Ciotoli et al. [23] in order to define the relationship between shallow distribution of gases and the main strike of the fault and verify whether the degassing process is still active along this fault trace. In fact, crustal discontinuities, such as fractures and faults of various dimensions, facilitate degassing flux from the Earth to the hydrosphere and the atmosphere. For this reason, the chemical composition and transport of soil gases within fault zones have been the subject of extensive investigations, including fault tracing and seismic surveillance as a precursor for geochemical anomalies to seismotectonic activity, due to their potential [24–35]. Faults can be described as weakened zones composed of highly fractured materials, gouge, and fluids. Active faults favor gas leaks because they usually increase the permeability of rocks and even their overlying soils. Gas anomalies at active faults can be either 'direct leak anomalies' where the gas measured corresponds to the deep gas phase or 'secondary anomalies' linked to the different mineralogical and hydrological behavior of the fault [29,36–38]. Anomalous gas emissions primarily occur in two different environments: Volcanic areas, where gas seepage is located both at central vents and often in large distal areas, and seismically active zones, where evidence of preferential degassing occurs near active faults. In the seismic zones, degassing has been shown to occur mainly as advective fluxes through soils of fractured areas and/or as free nonmixed gas phase from thermo-mineral springs due to pressure drop during ascent of the fluid to the surface [39]. The gas emission over seismically active faults corresponds to a long-term permanent phenomenon (with respect to earthquake recurrence times), which indicates that active faults are characterized by a high permeability and act as preferential conduits in the crust [39]. Several gases with different origins and contrasting behaviors in soil have been documented for detecting a fracture network and characterizing its extension and shape [37–40]. Because fluid transfer in the crust is strongly promoted by fractures, high geochemical contrasts are expected in faulted zones (e.g., [27]). The composition and distribution of gases in the soil pores are affected by surface features, such as pedological, biogenic, and meteorological factors. Several phenomena, like variation of the groundwater table, meteorological changes, soil porosity/permeability, and the

degree of fracturing, may alter original gas concentrations for a single gas. However, these are thought to have a subordinated effect on gas leakage from deep fault-related features [36,38,40], and can be further reduced by studying a number of soil gases with different origins and contrasting behaviors. In fact, even in a restricted fractured area, gas distribution in soil can display contrasting patterns. It has been shown that the contrasting permeability in fault gouges and intensely sheared zones generate complex geochemical patterns in soil atmospheres [40]. This characteristic has already been used to search for active faults mainly using Rn emanation, but sometimes also using $H_2$, He and $CO_2$ distribution in soils.

The idea that $CO_2$ overpressure controls earthquake nucleation in the Apennines [41] is consistent with large $CO_2$ and $CH_4$ emissions at surface [23,42], while the coexistence of $CO_2$ and brine is well documented in several wells [43]. The presence of liquid and gas fluid phases in a fault volume has important consequences on seismicity production. In fact, the presence of fluid within the fault gauge may enhance seismicity due to lubrication mechanisms. Seismicity may also be enhanced by an increase of pore pressure in the country rocks embedding the faults. In the Irpinia region, modelling of microearthquake spectra has provided a rather low average seismic radiation efficiency [44], thus implying that rupture lubrication mechanisms are not favored. Therefore, it may be envisaged that the dominant mechanism triggering and controlling microseismicity in the Irpinia region is the pore pressure increase induced by fluid diffusion in the host rock medium [45,46]. In particular, at hypocentral depths, gases may significantly increase the pore pressure compared to liquids, up to a level for which it equals the lithostatic pressure [47]. According to Amoroso et al. [8], the high-resolution 3-D P- and S-wave velocity models of the Irpinia fault zone highlight a significant fluid accumulation within a 15 km wide volume of highly fractured rock located between SW and NE boundary faults as indicated in Figure 2b. A high $V_P/V_S$ ratio recorded by seismic tomography points to the presence of fluid reservoirs below the mélange zone [8], consisting of brine-$CO_2$/$CH_4$ or $CO_2$-$CH_4$ mixtures [48]. The background micro-seismicity was therefore attributed to pore pressure changes in fluid-filled cracks surrounding major faults, which can trigger the episodic nucleation of moderate to large earthquakes. Analyses of micro-earthquake sequences revealed that they are primarily concentrated in densely fractured and limited regions characterized by horizontal, NE-SW-directed extension. These fault zones could be the source of repeated earthquakes due to the internal mechanical readjustments from local stress release and/or fluid migration along the fault damage zone [49].

## 2. Tectonic Framework

The southern Apennines form part of the Alpine-Apennine orogenic system, which derived from the convergence of the African and Eurasian plates in Late Cretaceous to Quaternary times (e.g., [50,51] and references therein). The Apennine accretionary wedge is composed of both ocean-derived [52,53] and continental margin-derived tectonic units. The latter include Mesozoic-Tertiary carbonate platform/slope successions (Apennine Platform) and pelagic basin successions (Lagonegro), stratigraphically covered by Neogene foredeep and wedge-top basin sediments (e.g., [54]) (Figure 2a). At the surface, the structure is characterized by low-angle tectonic contacts separating the Apennine Platform carbonates in the hanging wall and the Lagonegro Basin successions in the footwall [55]. These tectonic contacts consist of both thrusts—in part reactivated during extensional stages—and newly formed low-angle normal faults [56]. The Apennine accretionary wedge is tectonically superposed onto the buried Apulian Platform, which has a thickness of 6 km to 8 km and consists of a Mesozoic-Tertiary shallow-water carbonate succession, continuous with the outcropping in the foreland to the NE ([57] and references therein). The detachment between the allochthonous units and the buried Apulian Platform unit is marked by a mélange zone of variable thickness, locally reaching ca. 1500 m [58] (Figure 2b). The buried Apulian Platform is characterized by reverse-fault-related, open, long-wavelength folds that form the hydrocarbon traps for the significant oil discoveries in southern Italy [57]. Deep exploration wells also document the local occurrence of $CO_2$ gas caps in the top part of structural culminations made of fractured Apulian Platform carbonates,

while waters of variable salinity occur below the gas caps (where these are present) and along the sides of the culminations [43].

Geophysical evidence has shown that the crystalline basement is involved in deep-seated reverse faulting [59–61]. The associated deformation is represented by significant vertical offsets along steep reverse faults, with relatively limited horizontal displacements.

Crustal shortening ceased in the Middle Pleistocene (e.g., [62]), when NE-SW-oriented horizontal extension became dominant over the whole orogen. Extensional faults postdating and dissecting the thrust belt (e.g., [63]) are also responsible for the active tectonics and seismogenesis in the southern Apennines (e.g., [64–66]).

## 3. Geological Framework of the Study Area

The study area is dominated by the Mt. Marzano morphostructural high, with a maximum elevation of 1579 m. The Mt. Marzano massif is composed of a pile of slope facies limestones and dolostones more than 2000 m thick, spanning in age from the Late Triassic to the Early Miocene [67]. The carbonates of Mt. Marzano massif, as well as those of platform facies forming the backbone of the Picentini and Alburni Mts. massifs (located to the west and south of Mt. Marzano, respectively) and of minor ridges interposed between the different massifs, are related to the Apennine Platform. Surface and subsurface (San Gregorio Magno 1 and Contursi 1 deep wells [68,69]) data from the Mt. Marzano area show that the Apennine Platform carbonates are tectonically sandwiched between the underlying Lagonegro basin strata (outcropping to the north of the Mt. Marzano massif) and the overlying Upper Cretaceous-Burdigalian, basinal Parasicilide unit, representing the deformed distal portion of the Apulian foreland palaeomargin [70,71] and forming part of the Internal successions in Figure 2a. The latter are covered by Burdigalian-Langhian foredeep and wedge-top basin deposits (Figure 2) [70,72]. In the study area, wedge-top basin deposits of Pliocene age (i.e., Zanclean and Piacentian) [54,72,73] consist of marine sediments (clays, sands, and conglomerates), locally passing upward to lacustrine and fluvial deposits, outcropping on top of the Mt. Marzano massif and in the adjacent southern (Tanagro river valley) and northern topographic lows (Figure 2a). Coeval, clayey-to-shallow marine deposits (related to the Ofanto wedge-top basin) occupy the northernmost part of the study area, i.e., the large Ofanto River valley.

In the Mt. Marzano area, the occurrence of a structural high of the Apulian Platform below the outcropping Apennine Platform carbonates and the underlying Lagonegro basin strata has been evidenced by seismic reflection profiles and gravity data (Figure 2b) (e.g., [7,74–76]). Similar to the structural traps of the oil fields of the Basilicata region south of our study area [57], the Apulian Platform positive structure in the Mt. Marzano area appears to consist of a large inversion feature controlled by multiple reactivation of a major SW-dipping fault of probable Triassic original age. According to this interpretation, substantial structural relief of the Apulian Platform in this area resulted from Late Pliocene to Early Pleistocene shortening and inversion, having been only weakly modified by subsequent late Quaternary extension. The top of the Apulian Platform carbonates, which is overlain by thick deposits over than 600 m thick including Messinian evaporites and mélange units (probably involving Lower Pliocene clastics), lies at a depth of 3977 m in the San Gregorio Magno 1 well [68], whereas in the Contursi 1 well [69], the top of the mélange, is found at a depth of c. 3100 m.

The fold and thrust structure is dissected by extensional faults, which control continental depocentres such as those occupying the Tanagro River valley [77] and the Piani di Buccino (hereinafter, PB) and Pantano di San Gregorio Magno (hereinafter, PSGM) basins [78,79]. The Tanagro basin, which hosts several terraced alluvial units framed in the late Early Pleistocene to late Middle Pleistocene timespan [77], formed along a major WNW-ESE-striking, NE-dipping normal fault (Tanagro fault) [77,78]. This fault has recorded repeat activity over the Middle Pleistocene. The formation of the Tanagro basin predates those of the PB and PSGM (Figure 2), which occurred in the late part of the Middle Pleistocene, and of several minor basins within the Mt. Marzano ridge [7]. Within the Mt. Marzano massif, high-angle WNW-ESE-trending strike-slip faults and east-trending, NE-dipping

normal faults offset the Triassic-Miocene succession and the overlying Pliocene deposits [72]. Such faults have been sealed by several remnants of Pliocene–Early Pleistocene erosional surfaces molded in the top surface of the massif and graded down to the SSE, with an overall stair-like arrangement, from around 1500 m to 600–700 m [78]. The erosional surfaces, which evidence progressive surface lowering that has affected the Mt. Marzano topographic high in Pliocene–Early Pleistocene times, predate the extensional faulting, which has been active in the Mt. Marzano area since the Middle Pleistocene. Extensional tectonics activated major WNW-ESE trending, both NE and SW-dipping faults, which caused the formation of fault-bounded mountain fronts and of the previously mentioned intramontane basins filled with lacustrine and/or alluvial successions. Extensional faulting in the area spanning from Mt. Marzano to the Alburni Mts. has been active in late Quaternary times, as it is shown by widespread geomorphological-geological and paleoseismological evidence (discussed in the following sections).

## 4. Morphotectonic Features of the Mt. Marzano Area

The Mt. Marzano and surrounding area are characterized by widespread evidence of Quaternary extensional tectonics. Stratigraphic and morphotectonic evidence allows unravelling faulting chronology. This points to a progressive younging, from the SW to the NE, of vertical motion initiation over Middle to Late Pleistocene times in the area spanning from the Alburni Mts. mountain front to the Mt. Marzano massif. In the whole area, in addition, several faults show geomorphological/stratigraphical evidence for Late Pleistocene-Holocene activity.

The oldest Quaternary extensional tectonics was recorded in the Tanagro river valley, to the SW of the Mt. Marzano massif. The Tanagro river valley follows a Quaternary continental basin formed along a major WNW-ES-striking, NE-dipping normal fault, namely the Tanagro fault [77] shown in Figure 2c. Seismic reflection profiles have [80] highlighted the occurrence of several NE-dipping splays synthetic to the major NE-dipping Tanagro fault and of a horst-and-graben structure in the southern sector of the valley, i.e., across the Alburni Mts.–Mt. S. Giacomo–Mt. Marzano transect. The sedimentary record in the Tanagro valley basin provides evidence for repeated activity of the Tanagro fault over the Middle Pleistocene. This is shown by (i) the backtilting of early Middle Pleistocene alluvial fans in the SE area; (ii) SW-tilting of late Middle Pleistocene travertine-alluvial body and related terraces in the northern Tanagro valley—NTV (see Figure 3 for location of terraces in the NTV); and (iii) fluvial terraces located in the hanging wall block at the northern termination of the Tanagro Fault, pointing to renewed aggradation (following dissection of the travertine-alluvial body) in response to damming of the Sele-Tanagro rivers valley along the Tanagro fault [77].

Post-late Middle Pleistocene activity of the Tanagro fault has not yet been proven by either geomorphic or stratigraphic evidence. On the other hand, evidence for younger, Late Pleistocene-Holocene faulting characterizes the Caggiano fault that bounds the southern Tanagro valley toward the NE (location in Figure 2c). This is shown by the occurrence of a post-glacial midslope bedrock fault scarp running along the southern slope of Mt. S. Giacomo. Holocene historical activity of this fault is also evidenced by paleoseismological investigations [81]. Based on morphostratigraphical evidence, alluvial deposition in the NTV depocenter predates the formation of the PB and PSGM basins. In fact, the formation of the PB and PSGM interrupted S-flowing fluvial paths and related sedimentary (alluvial fan) inputs from the Mt. Marzano massif to the NTV, as it is inferred from the paleo-alluvial fan related to the Palomonte wind gap (Figure 3), which is cut into the PB southern border.

Surface and well data show that the PB and PSGM are filled with fine-grained deposits (mainly sandy-silts and clays), passing laterally into alluvial fans and slope debris. In those basins, lacustrine/marshy environments were present in historical times and persisted until they were reclaimed in the XX century [79]. The PB and PSGM basins are bounded toward the NE by a major WNW–ESE-trending fault zone consisting of N100° to N120° en-echelon normal faults, N70°-striking transfer faults (probably reactivating pre-existing inherited structures) [7], and minor antithetic faults toward the SW. The occurrence of a major northern fault zone is consistent with the N-ward thickening of the PB fill up to about 100 m [7]. The thickness of the PSGM basin fill, which exceeds the 62 m depth of a core drilled at the basin axis, is not well constrained. The carbonate bedrock becomes shallower (about 35 m deep) toward the eastern basin termination, as shown by geophysical investigations [82]. Geophysical investigations have also imaged the WNW-ESE-trending northern and southern (i.e., the NE-dipping Mt. Difesa Ripa Rossa; location in Figure 3) normal faults down to a depth of c. 300 m. A lateral variation in the amount of offset along the PSGM northern fault zone is made evident by the occurrence of a bedrock high between the PSGM and PB basins.

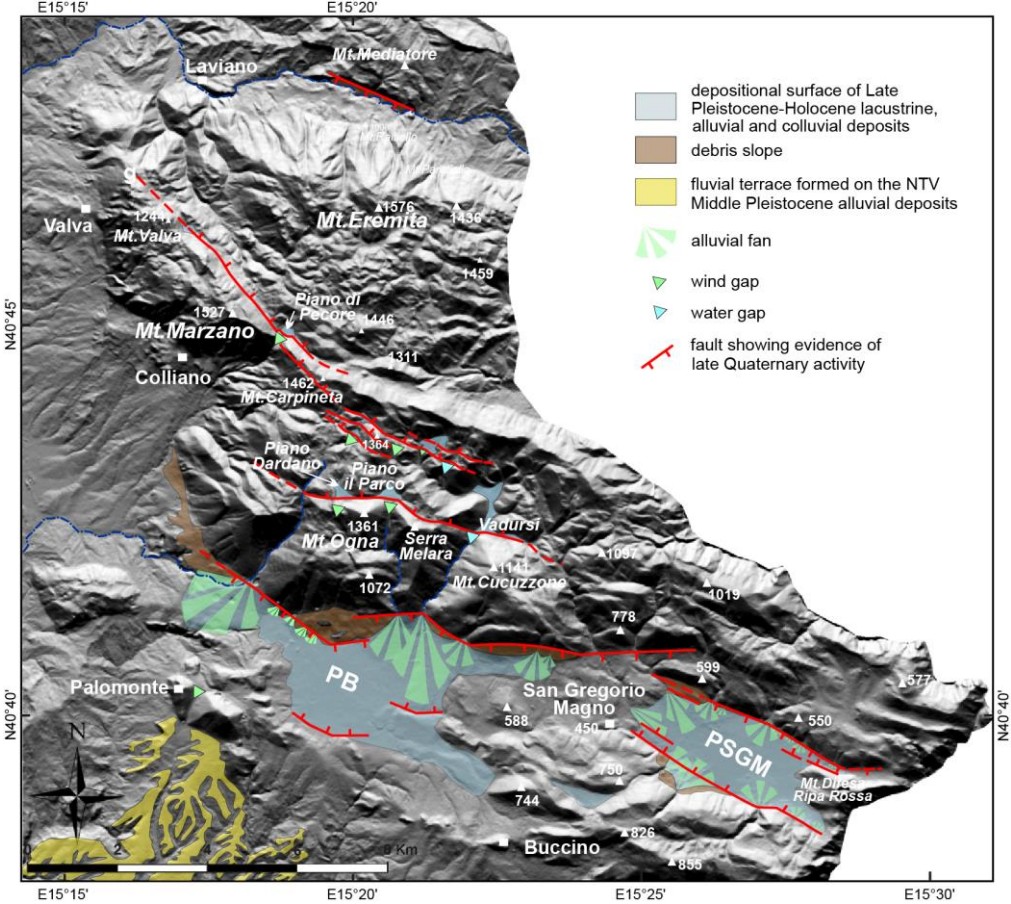

**Figure 3.** Faults showing evidence of late Quaternary activity in the Mt. Marzano ridge area (redrawn after Ascione et al. [7]).

Scraps around a few meters high affect Holocene lacustrine terraces and alluvial fans within the PSGM and PB basins. Furthermore, bedrock fault scarps and wineglass valley cross-profiles, as well as faulted Upper Pleistocene-Holocene slope breccia and alluvial fan deposits, occur along the entire northern fault-bounded mountain front. To the east, a debris slope is offset along the Balvano fault. The heights and estimated ages of both bedrock and alluvial fault scarp along the northern fault bounded mountain front of both PB and PSGM basins suggest that fault slip rate only slightly exceeds the accumulation rate within the two basins (ranging from 0.1 to 0.37 mm/a, with a mean value of 0.2 mm/a in the last 240 ka in the PSGM [7]). The slip rate estimated so far is consistent with the value (<0.5 mm/a) independently estimated by slip profiles analysis of fault scarps in the northern mountain front [20].

Evidence for Late Pleistocene-Holocene fault activity in the Mt. Marzano massif is widespread further to the north and east of the PB-PSGM major fault (Figure 2c). Among the faults active since the final part of the Middle or the Late Pleistocene are the NE-dipping Mt. Valva–Mt. Marzano–Mt. Carpineta and the Mt. Ogna–Mt. Cucuzzone fault zones, consisting of a succession of fault strands with average NW-SE to WNW-ESE strike (Figure 3). Offsets accumulated by these younger faults have not significantly affected topography of the Mt. Marzano massif, which culminates on the Mt. Eremita peak (1579 m) located in the northernmost part of the massif. Recent activity of NE-dipping faults in the Mt. Marzano ridge have interacted with surface processes, causing widespread valley damming (evidenced by several wind gaps) and the formation of alluvial basins (Figures 3 and 4). Recent activity has also affected the longitudinal profiles of the streams that dissect the ridge (Ascione et al. 2013).

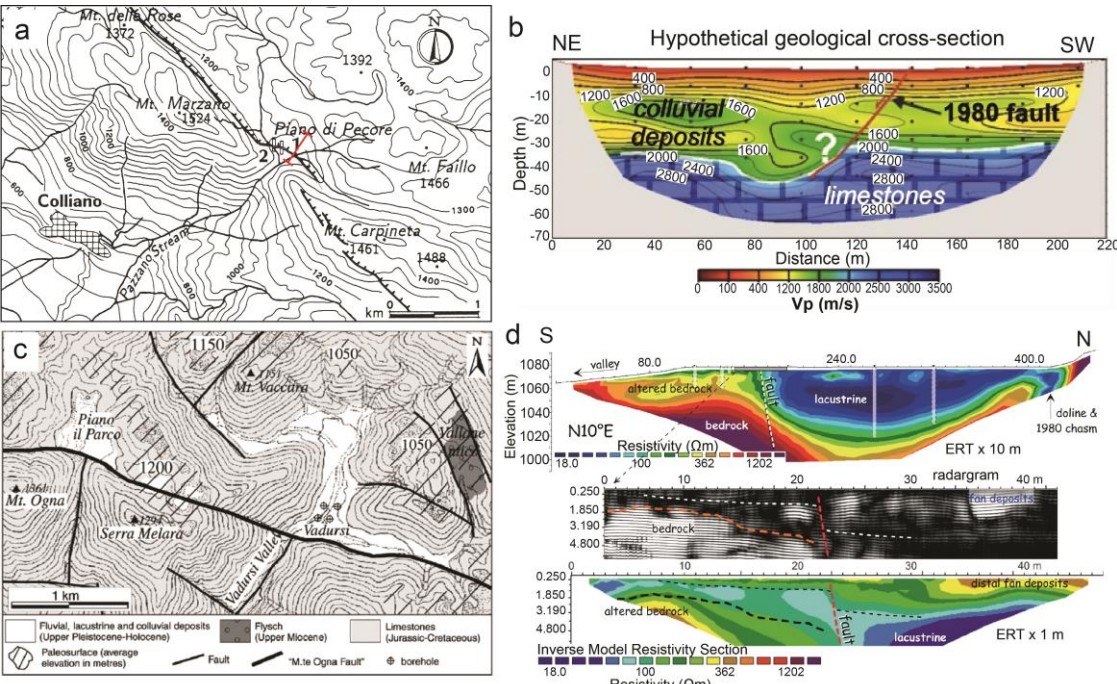

**Figure 4.** (**a**) Location of the Piano di Pecore basin, created by fault damming of the Pazzano stream with an indication of the coseismic N-facing fault scarp (from Pantosti et al. [13]). (**b**) Seismic tomography and inferred geological cross-section across the Piano di Pecore basin, along the red trace in diagram a, showing thickening of the basin fill toward the southern basin border (from Ascione et al. [78], modified). (**c**) Geological sketch map of the Mt. Ogna area, with the Piano il Parco and Vadursi basins, created by damming of S-flowing stream valleys (from Ascione et al. [78], modified). (**d**) Geophysical investigations of the Piano il Parco basin, showing thickening of the basin fill toward the southern basin border (from Galli et al. [83], modified).

Due to the carbonate nature of the rocks that form the backbone of the Mt. Marzano ridge, the dammed valleys evolved as karst basins in several instances. One of such basins is occupied by the Piano di Pecore plain. The Piano di Pecore basin is an undissected karst basin filled with continental deposits with an inferred thickness around 30–40 m (Figure 4). The basin fill, based on surface evidence and on shallow excavations (paleoseismological trenches, see Section 6), is composed of slope debris, limited to the basin margins, as well as fine-grained colluvial, lacustrine-marshy, and volcanoclastic sediments. The formation of the Piano di Pecore basin occurred in response to the damming of a S-flowing stream (Pazzano stream) by the NE-dipping Mt. Valva–Mt. Marzano–Mt. Carpineta fault zone. This is inferred from the occurrence, at the SW border of the basin, of a 200 m deep wind gap cut across the Mt. Marzano-Mt. Carpineta ridge (Figure 3) [78]. Evidence for recent (Holocene) activity of the Mt. Marzano–Mt. Carpineta fault is provided by the 5 m vertical separation between the undissected basin floor and the wind gap bottom, whereas the youngest tectonic activity has been evidenced by the 1980 coseismic rupture (see Section 6). Tomographic investigations allowed the identification of the fault activated with the 1980 earthquake in the Piano di Pecore subsurface by a bedrock throw of approximately 20 m, which affects the top of the carbonate bedrock of the basin (Figure 4). The bedrock throw has most probably overestimated the total fault offset which, based on the vertical separation between the basin bottom and the adjacent wind gap bottom, has been evaluated at about 40 m, consistent with the 40–50 m vertical separation between displaced erosional surface cut on both the hanging wall and the footwall blocks of the entire Mt. Valva–Mt. Marzano–Mt. Carpineta fault zone [78].

## 5. Seismicity

Active tectonics evidence the intense seismicity affecting the Mt. Marzano area, which falls within the epicentral area of some of the strongest historical earthquakes of southern Italy, i.e., those with intensity I ≥ X MCS occurred in 989, 1694, in 1930, 1962 and 1980 (Figure 2a) (e.g., [11,14–16,84]). Among such events, the $M_S$ = 6.9 1980 Irpinia earthquake dramatically struck the Mt. Marzano and the neighboring area, which recorded widespread coseismic ruptures and a large number of secondary geological effects, including landslides, ground cracks, liquefaction, and variations in the discharge rate of major carbonate springs [14]. The area, which was also struck by a seismic sequence with a $M_L$ = 4.9 mainshock [85] in 1996, is presently affected by subdued background seismicity with $M_L$ ranging between 1 and 3.3 (Figure 5) [4,86–88]. In July 2020, the NW termination of the extensional fault system (in the Rocca San Felice area; location in Figure 2a) was affected by a minor seismic sequence (with maximum $M_L$ = 3.0 recorded for two shocks), including 43 events distributed over a NW-SE-trending, narrow, ca. 5 km long zone [87]. Using the 3D velocity model described by Amoroso et al. [8], 36 selected events were relocated, depicting a fault zone dipping 50–60° toward the NE [87]. The related focal mechanisms show fault plane solutions characterized by dominate dip-slip, normal fault kinematics with a subordinate strike-slip component. The seismological dataset associated with this seismic sequence, being generally consistent with the seismogenic sources of the Irpinia region, has recorded fault activity in the proximity of the NW tip of the regional seismogenic structure. One of the largest nonvolcanic natural emissions of low-temperature $CO_2$ rich gases ever measured on Earth, i.e., the Mefite d'Ansanto (e.g., [89]), is located in this area. This is consistent with previous studies suggesting that the densely fractured 'process zone' surrounding fault tips represent major conduits for fluid flow, giving rise to strong gas emissions at the surface [23]. Boreholes in this area also indicate the occurrence of a $CO_2$ gas cap at the top of an antiformal trap involving fractured Apulian carbonate reservoir rocks, while saline water variably occurs along the sides of the structural trap and below the gas cap [43].

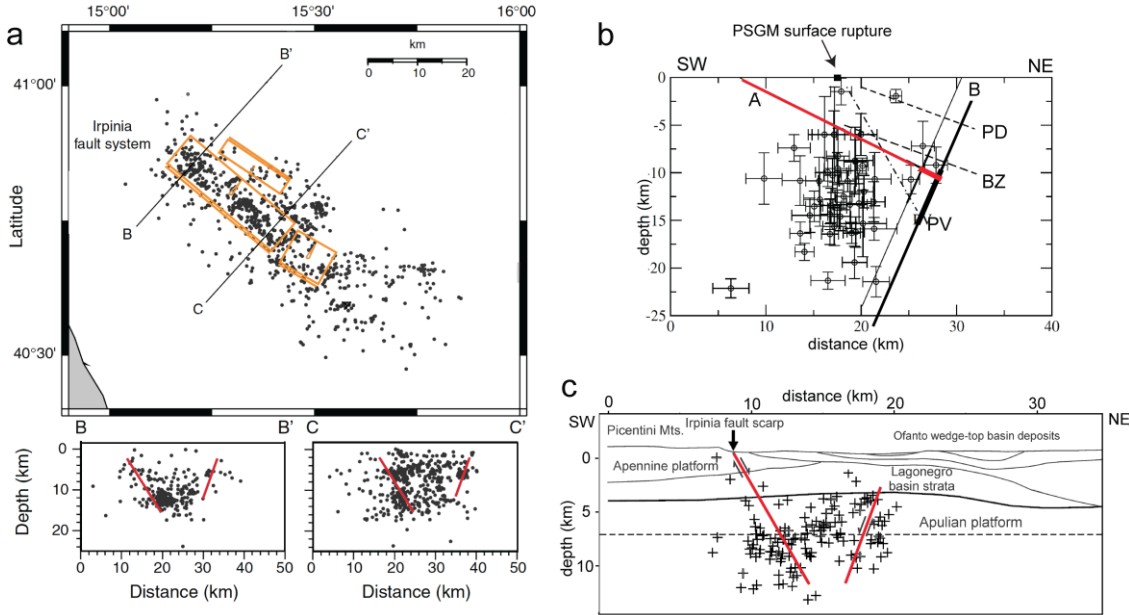

**Figure 5.** (**a**) Seismicity of the Irpinia region from August 2005 to April 2011 in map view (in orange, seismogenic sources from DISS working group [88]) and in cross-sections, along the profiles reported in the map (the red lines represent the projection of the fault segments activated with the 1980 earthquake) (from De Matteis et al. [4], modified). (**b**) Comparison of models for the 20 s fault, and projection of aftershocks (crosses indicate uncertainty) localized on a vertical N31E cross-section cutting across the PSGM; A and B (effectively slipped area: thick line): Amoruso et al. [90]; BZ: Bernard and Zollo [1]; PD: Pingue and De Natale [91]; PV: Pantosti and Valensise [12] (from Amoruso et al. [90], modified). (**c**) Gravity profile, and the fault segments and aftershocks [92] (crosses indicate uncertainty) of the 1980 earthquake (partly redrawn from Improta et al. [74]).

## 6. The 1980 Irpinia Earthquake

As already mentioned above, the 1980 Irpinia earthquake was characterized by a complex source mechanism, consisting of three major subevents at 0 s, 20 s and 40 s, with $M_w$ 6.9, 6.4, and 6.3 respectively (Figure 1) (e.g., [1,3]). The mainshock originated in the 8–13 km depth range [93]. Bernard and Zollo [1] defined the three normal faulting ruptures as follows: (i) The $M_S = 6.9$ mainshock, which nucleated on the NE-dipping Mt. Marzano and Picentini Mts. segments (about 20 km long); (ii) the 20 s subevent, which nucleated about 15 km southwest of the first event on a circa 20 km long normal fault; (iii) the 40 s subevent, located in the Ofanto basin area, which was associated with a SW-dipping normal fault antithetic to the first activated fault (Figures 1B and 5). According to Nostro et al. [94], the 40 s subevent might have reactivated a fault segment which ruptured during the Me = 6.9 [95], 1694 earthquake, which struck an area overlapping with that hit by the 1980 event (e.g., [14,15]). Pingue and De Natale [91] estimated an 80° SW dip for the Ofanto fault. On the other hand, the NE-dipping fault segment responsible for the Mt. Marzano mainshock is constrained in a 53°–63° dip range by seismological data [93], whereas dip values of 60° (at depth) to 70° (at surface) have been proposed based on breakout and log analysis of the S. Gregorio Magno 1 well [96]. Probably due to the scarce source coverage [97], the localization and mechanism of the 20 s subevent are debated. In fact, the 20 s nucleation has been associated with: (i) A deep-seated, NE-dipping, low angle (20°) fault [1]; (ii) a fault plane dipping 60° to the NE [12]; and (iii) a SW-dipping fault antithetic to that of the mainshock (and roughly aligned with the 40 s fault; Figure 5b), which is suggested to represent a reactivation of the southernmost part of the fault activated with the 1694 earthquake [90]. On the other hand, based on absence of record of the 1694 event in trenches dug along the 1980 ruptures, it has been proposed that the fault responsible for the 1694 earthquake was similar to and longer than the 40 s fault [98].

However, in the paleoseismological trenches (see below), no record of other large earthquakes that struck the area, e.g., those that occurred in 1930 and 1962, was found.

The Mt. Marzano–Picentini Mts. fault segments and the antithetic Ofanto fault define the boundaries of both a NW-trending zone of aftershocks (which coincide with a high P-wave velocity zone) [92,99] and a volume affected by background microseismicity on subparallel predominantly normal faults (Figure 5a) [4]. The 1980 earthquake affected the epicentral area with widespread coseismic surface ruptures and ground deformation, which have been detected by geodetic levelling surveys [1,22,91].

Post-eventum surveys in the region recording the largest intensity ($I_0 \geq IX$) have reported more than 3000 ruptures, with those with a vertical offset showing a marked N120° main trend [100]. Among such ruptures, the circa 2 km long, up to 50 cm high, WNW-ESE trending, NE-facing scarp formed in the PSGM basin (Mt. Difesa Ripa Rossa scarp [19]; location in Figure 3) and the mainly NW-SE trending, NE-facing scarps in the Mt. Marzano–Piano di Pecore area [18] were interpreted as surface faulting (Figure 6).

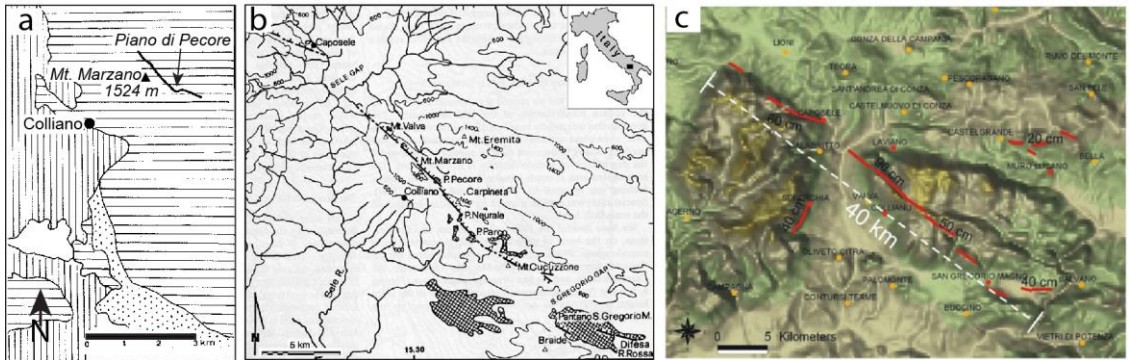

**Figure 6.** (**a**) Post-eventum mapping of the 1980 Irpinia earthquake surface faulting in the Mt. Marzano peak—Piano di Pecore plain area (modified from Cinque et al. [18]). (**b**) Mapping of the 1980 Irpinia earthquake coseismic surface ruptures from [12]. (**c**) Distribution of the 1980 Irpinia earthquake coseismic surface faulting, after [10,12,14,18,19,101,102] (from Serva et al. [15], modified).

In particular, a high scarp ranging from several tens of centimeters to around 1 m, running across the NE slope of the Mt. Valva–Mt. Marzano ridge and the floor of the Piano di Pecore plain and continuing toward the SE for some hundreds of meters, was recognized [18]. The scarp was characterized, for most of its length, by a NW-SE trend and a NE dip. A roughly E-W trending, N-facing scarp was also recognized (Figure 6). The latter scarps, and those affecting the SW slope of Mt. Carpineta ridge (the downthrown block being the uphill side), were interpreted as part of a 10 km long succession of right-stepping strands of the 1980 surface faulting, and the PSGM scarp was related to a further fault segment [8]. Coseismic faulting was related to three main strands separated by gaps (i.e., the Sele valley and San Gregorio Magno gaps), with changes in strike of about 10° [12]. Such fault strands, interpreted as part of a 38 km long, NE-dipping scarp with an average N128° strike ('Irpinia fault'), are identified with: (i) The Picentini Mts. footslope, with a N125° trend; (ii) the N135° trending Mt. Marzano–Mt. Valva scarp coupled with the N135°–N110° trending Mt. Carpineta scarp; and (iii) the N130°–N120°-striking PSGM scarp (Figure 5). The Mt. Marzano–Mt. Valva–Mt. Carpineta and Picentini Mts. segments have been associated with the mainshock and the PSGM strand with the 20 s subevent. It has also been suggested that the Mt. Valva–Mt. Marzano–Mt. Carpineta and the Picentini Mts. strands are part of single, continuous fault having a subdued topographic expression as it crosses the Sele River Valley, which is occupied by rocks with a low resistance to erosion [20]. The 40 s subevent has been associated [101] with circa 8 km long, up to 30 cm high, SW-facing scarps exposed between Santomenna and Muro Lucano (Figure 6c). Paleoseismological investigations carried

out in the Piano di Pecore site revealed a mean slip rate around 0.3 mm/a [13], and a 0.17–0.4 mm/a slip rate in the last 20 ky was estimated for the antithetic, NE-dipping, Mt. Difesa Rossa fault [103].

Recently, Ascione et al. [7] identified several fault arrays within the epicentral area of the 1980 earthquake (Figure 2c). Within the structures belonging to the identified fault arrays, those developed in the area spanning from the northern Mt. Marzano massif (i.e., the Marzano fault array) to the Ofanto river valley—being characterized by a NE dip in the Mt. Marzano area and a SW dip to the NE of such a massif—are considered as the surface expression of the deep-seated graben-like structure which was activated with the 1980 earthquake (e.g., [1,90,91]). Their spatial distribution overlaps the belt affected by the 1980 earthquake aftershocks (e.g., [92]) and by the present-day low magnitude seismicity, which occurs on subparallel, predominantly normal faults (Figure 2e) [4]. A major SW-dipping active fault system was identified to the SW of the earthquake epicenter, at the northern boundary of the S. Gregorio Magno—Buccino basins (Figure 2c) [7]. Stress inversion from surface faults and from instrumental earthquake focal mechanisms show a consistent pattern of NE-SW roughly horizontal maximum extension (Figure 2d), which is compatible with the T axis obtained from the 1980 main shock and with results of cumulative stress inversion from earthquake focal mechanism data (Figure 2e). Surface structures are decoupled from the seismically active deep-seated structures, which, in turn, reactivate inherited basement faults (Figure 2b) [7]. Decoupling between deep and shallow structural levels is related to the strong rheological contrast produced by the fluid-saturated, clay-rich mélange zone interposed between the foreland Apulian Platform carbonates and the allochthonous units, which include carbonate platform (i.e., Apennine Platform) and basin (i.e., Internal and Lagonegro) successions (Figure 2b). Based on such an interpretation, the surface expression of the deep SW Boundary Fault (SWBF) is represented by the PB-PSGM fault zone, that of the Central Fault (CF) by the Mount Marzano–Mount Carpineta (MC) and Mount Ogna (Og) fault zones and that of the NE Boundary Fault (NEBF) by the Conza (Co) and S. Fele fault strands (Figure 2b,c).

Amoroso et al. [8], thanks to the installation of dense, high dynamic range, seismic network operated by INGV (Istituto Nazionale di Geofisica e Vulcanologia) and AMRA (Analisi e Monitoraggio dei Rischi Ambientali) in the area struck by the 1980 Irpinia earthquake, recorded a massive waveform dataset of micro-earthquakes with magnitude larger than about 1 from August 2005 through April 2011. The researchers analyzed $V_P$ and $V_S$ wave velocities in the upper crust of the Irpinia fault system and the related microseismicity distribution, which appears to be confined within an uplifted block including the main normal fault rupture of the 1980 earthquake (Figure 7). They observed high $V_P/V_S$ and low $V_P \times V_S$ values in the region where intense microseismicity is located, which suggests fluid accumulation within a ~15 km wide rock volume. Further studies [48] have provided evidence for a composition of the fluids permeating the subsurface of the 1980 Irpinia earthquake region dominated by $CO_2$ and brine.

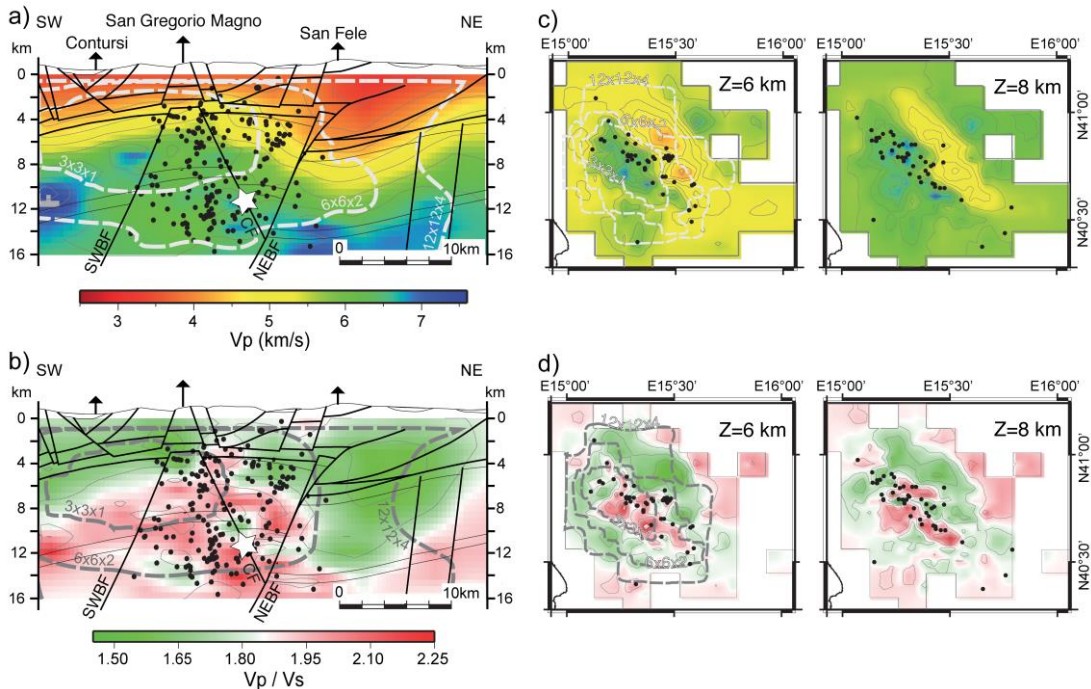

**Figure 7.** Three-dimensional tomographic model of the 1980 Irpinia earthquake region. (**a**) $V_P$ velocity model and microearthquake locations projected onto the cross-section located in Figure 2a (refer to Figure 2b for the tectonic contacts and geological units). Dashed curves delimit the well-resolved regions of the model. Each curve corresponds to a different resolution scale obtained from estimating the resolvability function for each model parametrization used in the multiscale approach, as explained in detail by Amoroso et al. [8]. Star shows the hypocenter of the $M_S$ 6.9, 1980 Irpinia earthquake. (**b**) $V_P/V_S$ ratio for the same depth section as in (a). (**c**) Horizontal slice through the P-wave tomographic model at depths (Z) of 6 km and 8 km. (**d**) Horizontal slice showing $V_P/V_S$ ratio at depths (Z) of 6 km and 8 km (modified from Amoroso et al. [8]).

## 7. Postseismic Deformation

Recent studies suggest that a significant proportion of the topography and geomorphology in tectonically active areas is controlled by deformation in the postseismic time period following earthquakes [104–107]. The deformation during the parts of the earthquake cycle that do not involve slip during earthquakes may be analyzed by means of satellite data to map the motion of the Earth's surface (known as space-based geodetic measurements, e.g., GPS and InSAR). At present, the length of the earthquake cycle (decades to millennia) rules out examining an entire cycle on a given fault using these space-based geodetic methods. However, even imaging parts of the coseismic, postseismic, and interseismic periods—using the decades-long archive that is now available for exploitation—may provide useful insights into the geometry and mechanisms of the deformation. Within this framework, it is crucial to compare the strain in the earthquake cycle—including that produced by postseismic afterslip—with the geology and topography of the study area by means of tectonic geomorphology analyses, which clearly provide insights into the deformation integrated over much longer timescales with respect to a single earthquake cycle.

*Construction of Ground Deformation Maps Based on PSs Mean Vertical Velocity*

In this section, we present the procedure followed to obtain two radar vertical mean velocity ground deformation maps that cover the region around the epicentral area of the 1980, 6.9 Mw, Irpinia earthquake. The maps were constructed using Permanent Scatterers—PS datasets processed by images recorded along both the ascending and descending orbits by ERS 1/2 (1992–2000) and Envisat (2003–2010) satellites. The PS datasets analyzed in this study are listed in Tables 1 and 2.

**Table 1.** ERS Permanent Scatterers datasets used in this study.

| PS ERS Original Dataset | Number of Pss |
| --- | --- |
| ERS_T358_F819_CL003_CAPOSELE_A | 38,864 |
| ERS_T358_F819_CL002_BENEVENTO_A | 45,672 |
| ERS_T494_F2781_CL001_POTENZA_D | 75,178 |

**Table 2.** ENVISAT Permanent Scatterers datasets used in this study.

| PS ENVISAT Original Datasets | Number of PSs |
| --- | --- |
| ENVISAT_T86_F816_CL001_FOGGIA_A | 391,518 |
| ENVISAT_T358_F801_CL001_SALERNO_A | 248,986 |
| PST2009_ENVISAT_T358_F819_CL001_BENEVENTO_A | 532,482 |
| ENVISAT_T265_F2781_CL001_AVELLINO_D | 463,801 |

The datasets listed in Tables 1 and 2 were obtained from the Geoportale Nazionale of the Italian Ministry of Environment (MATTM) database (www.pcn.minambiente.it/mattm) and preprocessed with PSInSAR by the TRE Company and PSP-DIFSAR by the e-GEOS Company. Details on the signal processing have been described by the authors of [108,109]. The PS position values for C-band SAR satellites, such as the ERS and ENVISAT satellites, may be affected by a measurement error of ±3 mm. However, the ERS and ENVISAT PS data from the MATTM database are of high quality. Any single pixel in a radar image is selected only if its temporal and geometrical decorrelation values are extremely low. In fact, the coherence values (ranging from 0 to 1) of the PS of the datasets that cover the investigated area are ≥0.6, which means that the PS coherence values are high. In other words, the analyzed PS are stable with respect to the radar and, thus, their motions are measured with high precision [109,110]. Consistently, the standard deviations of mean velocity values of single PS, calculated over the timespans of the entire records (i.e., the 1991–2000 and 2003–2010 timespans, respectively), are ≤0.5 mm/y for about the 90% of both datasets.

A GIS geospatial data analysis was used to construct, by means Arcgis 10.7® software (ESRI, Redlands, CA, USA), a series of raster maps synthesizing the spatial distribution of various parameters at different timespans. For the geospatial analysis, the Inverse Distance Weighting interpolator (IDW) statistics were used. In the first steps, the entire original ERS and ENVISAT PS datasets (listed in Tables 1 and 2) were processed. Afterward, the PS datasets were analyzed in order to select, from the original datasets, subsets of data (hereinafter labelled PS normal subsets) by excluding outlier data (see below).

Due to the near-polar orientation of SAR orbits, components of ground-motion-oriented N-S are substantially undetectable, while E-W-oriented components contribute to the motion detected by SAR satellites along its Line of Sight (LoS). Starting from PS motion recorded along SAR both ascending and descending orbits, the displacement/velocity in the vertical plane (z) oriented E-W may be reconstructed through the evaluation of the vertical (Dz or Vz) and horizontal (E-oriented, Deast or Veast) components of PS displacement, or velocity using the following equations by the authors of [111–113]:

$$Dz = (D_{LoSd} + D_{LoSa})/2\cos\phi \tag{1}$$

$$Deast = (D_{LoSd} - D_{Losa})/2\operatorname{sen}\phi \tag{2}$$

$$Vz = (V_{LoSd} + V_{LoSa})/2\cos\phi \tag{3}$$

$$Veast = (V_{LoSd} - V_{Losa})/2\operatorname{sen}\phi \tag{4}$$

where $D_{LoSd}$ and $D_{LoSa}$ are PS displacement values oriented along the descending and ascending LoS, respectively; $V_{LoSd}$ and $V_{LoSa}$ are PS velocity values oriented along the descending and ascending LoS, respectively; and $\phi$ is the LoS angle of incidence, which is around 23° for ERS and ENVISAT satellites.

The SAR data analysis was focused on a key area including the Mt. Marzano and Conza sites, where surface evidence of the main NE-dipping fault responsible for the 1980 Irpinia earthquake mainshock and the antithetic SW-dipping fault responsible for the 40-s subevent occurred [7,22]. The perimeters of the areas of interest (shown in Supplementary Figure S1) were traced along the regions where the ascending and the descending original datasets and subsets are consistently overlapped. This is a necessary condition to operate, subsequently, on the interpolated raster data in order to derive the horizontal and vertical components of the ground motions by applying the relationship [3]. As a consequence, the parameters of the analyzed ERS and ENVISAT datasets are slightly different (Supplementary Figure S2).

The IDW analysis, with cell size 50 × 50 m, was applied to the ERS PS 'native' ascending and descending datasets (i.e., the datasets that include both normal and outlier PSs) in order to construct the LoS-oriented mean velocity maps, which are reported in Supplementary Figure S3. Afterward, starting from the maps of the ascending ($V_{LoSa}$) and descending ($V_{LoSd}$) mean velocities, by applying Equation [3], the vertical component of mean velocity (Vz), or ground deformation, was constructed for the 1992–2000 timespan (Figure 8).

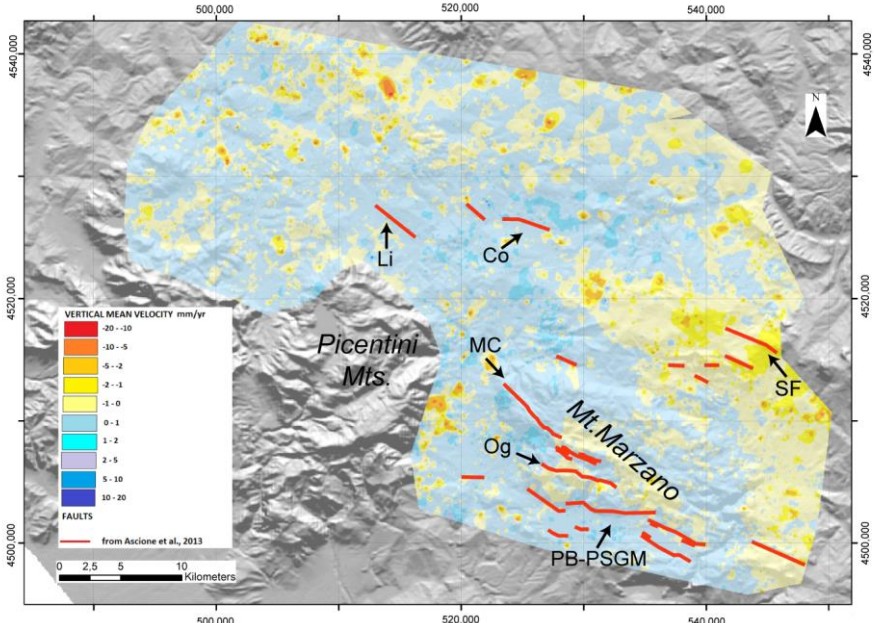

**Figure 8.** IDW interpolation (cell size 50 × 50 m) of ERS PS 'native' datasets (i.e., the datasets that include both normal and outlier PSs), showing the vertical mean velocity/ground deformation in the 1992–2000 timespan. Fault/fault array labelling as in Figure 2b.

The maps of the LoS-oriented (ascending and descending) mean velocities in the 2003–2010 timespan, constructed using the 'native' ENVISAT datasets, are reported in Supplementary Figure S4. Through the relationship [3], the vertical component of mean velocity for the ENVISAT PS original dataset was obtained (Figure 9).

A geospatial analysis was then carried out by applying both the *Hot Spot Analysis* (Getis-Ord Gi*) and the *Cluster and Outlier Analysis* (Anselin Local Moran's I) mapping tools to all the PS subsets without applying any filter based on coherence values. The data analysis started with the extraction of 'PS normal subsets' from the 'native' datasets through the outliers' boundaries (fence) evaluation (Supplementary Figure S5 and Table S1). Afterward, the PSs (of both the ERS and ENVISAT datasets) falling inside the perimeters shown in Supplementary Figure S2 were selected. The data included in the investigated perimeters form three new subsets for the ERS data, and four new subsets for the ENVISAT data, all identified with the label '*marzano'* (Supplementary Figures S6 and S7).

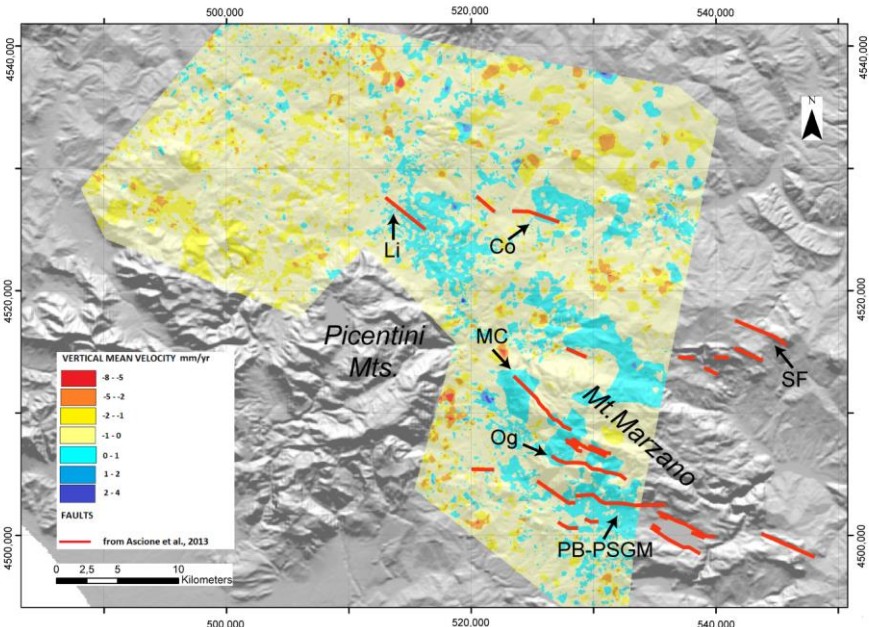

**Figure 9.** IDW interpolation (cell 50 × 50 m) of ENVISAT PS 'native' datasets (i.e., the datasets that include both normal and outlier PSs), showing the vertical mean velocity/ground deformation in the 2003–2010 timespan. Fault/fault array labelling as in Figure 2b.

Following, in part, the procedure by [114], both the *Hot Spot Analysis* (Getis-Ord Gi*) and the *Cluster and Outlier Analysis* (Anselin Local Moran's I) mapping tools for all 'marzano' were applied to the PS subsets (the subsets arising from the *Hot Spot Analysis* are labelled with 'HS,' and those from the *Cluster and Outlier Analysis* with 'CO'). The label '2k' of the subsets' names indicates the 2000 m *Threshold Distance* that is '*a cutoff distance for Inverse Distance option. Features outside the specified cutoff for a target feature are ignored in analyses for that feature.*' The label 'IDW' is referred to the Conceptualization of Spatial Relationships that is '*nearby neighbouring features have a larger influence on the computations for a target feature than features that are far away.*' The mapping results are shown in Supplementary Figures S8 and S9 for both of the used tools. Inspecting the diagrams in Supplementary Figures S8 and S9, it is evident that both mapping tools give a similar distribution of representative PSs (the red and blue points), and there is a better data visualization in the maps created with the *Cluster and Outlier Analysis* (CO maps). For these reasons, the CO maps were used for any subsequent elaboration.

To operate with a less but still significant number of PSs, we further removed, from the 'CO' subsets, the PSs defined as 'not significant,' 'high outlier' (HL), and 'low outlier' (LH). Examples of maps constructed using ERS and ENVISAT descending subsets composed of PSs classified as 'not significant' (nsig) are shown in Supplementary Figures S10 and S11. The remaining PSs, classified as (HH) and (LL), were selected for the new subsets (identified by the 'nout' label; Supplementary Tables S2–S5), which were used to construct new ground deformation maps for the investigated area (see below). Applying the IDW interpolation (cell size 50 × 50 m) to data of the PS subsets labelled 'CO_2k_IDW_nout', maps of the LoS-oriented ascending and descending mean velocities ($V_{LoSa}$ and $V_{LoSd}$) were constructed (Supplementary Figures S12 and S13). Afterward, Equation (3) was applied to both the ERS and ENVISAT LoS-oriented ascending and descending velocities to construct maps of the vertical component of mean velocity/ground deformation in the 1992–2000 and 2003–2010 timespans, which are shown in Figures 10 and 11, respectively.

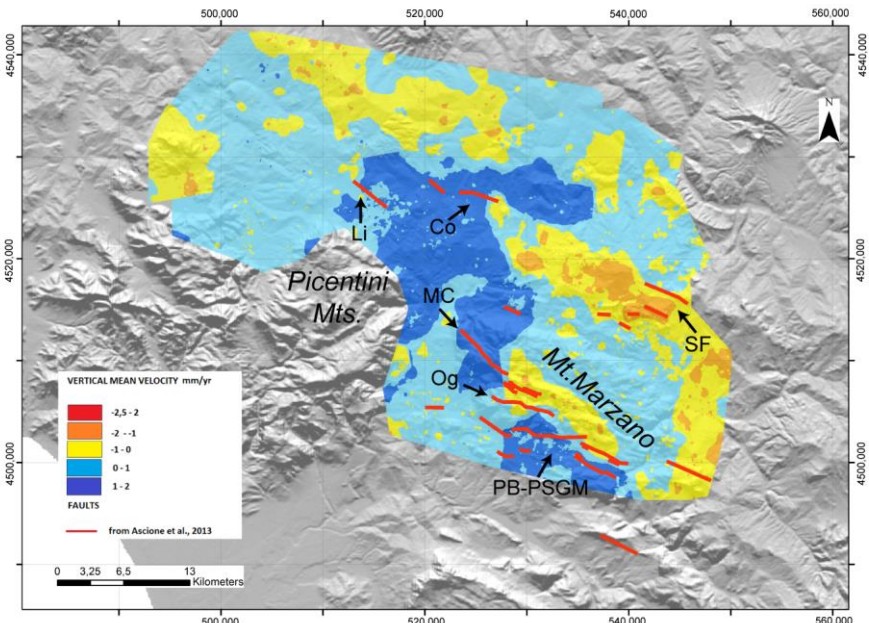

**Figure 10.** ERS data: Vertical mean velocity/ground deformation map 1992–2000, IDW interpolation (cell size 50 × 50 m). Data statistical selection with the *Cluster and Outlier Analysis*, i.e., map constructed using the ERS subsets labelled 'CO_2k_IDW_nout' (which includes only PS classified as HH and LL). Fault/fault array labelling as in Figure 2b.

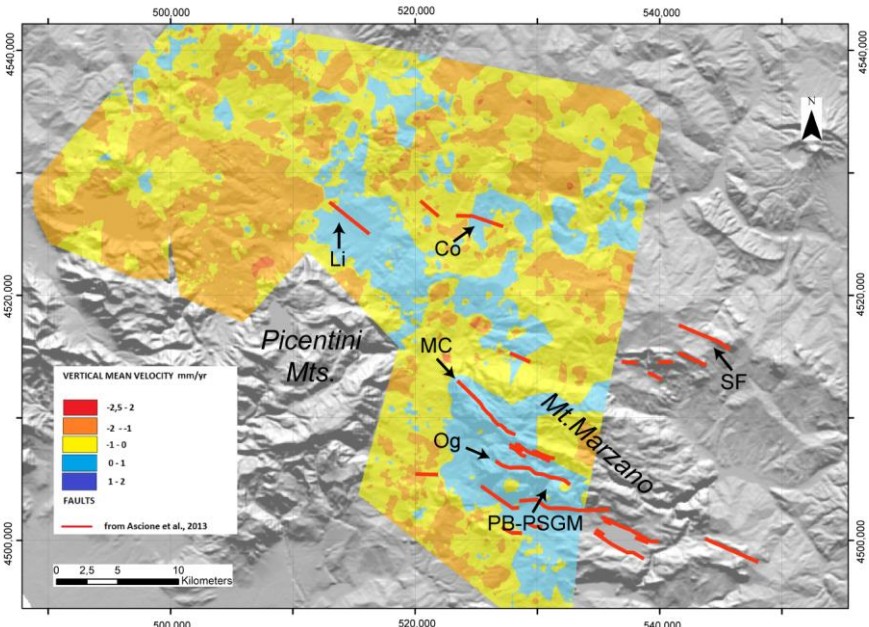

**Figure 11.** ENVISAT data: Vertical mean velocity deformation map 2003–2010, IDW interpolation (cell 50 × 50 m). Data statistical selection with *Cluster and Outlier Analysis*, i.e., map constructed using the ENVISAT subsets labelled 'CO_2k_IDW_nout' (which includes only PS classified as HH and LL). Fault/fault array labelling as in Figure 2b.

## 8. Discussion of Postseismic Deformation

Two types of processing were used in this study. One technique (i.e., IDW analysis of 'native' PS datasets) requires few steps of analysis, whereas the other one (based on the *Cluster and Outlier Analysis*) involves a more complex sequence of steps. The maps obtained by means of these two types of processing are quite similar in terms of spatial distribution of vertical (up/down) motion orientations (Figures 8–11).

Both ERS maps show that the area of interest, is for the most, part slightly uplifted (0–1 mm/y mean velocity class). However, the ERS vertical mean velocity deformation map for the timespan of 1992–2000, obtained with the *Cluster and Outlier Analysis* (CO map of Figure 10), shows the presence of areas characterized by slightly higher positive velocity (1–2 mm/y mean velocity class) in the 1992–2000 timespan more impressively than the map of Figure 8. These areas are located in the areas corresponding to the surface projections of the footwall blocks to the major blind seismogenic faults identified as (i) Central Fault (CF) (particularly, the northern and southern parts of such block) and (ii) NE Boundary Fault (NEBF) [8]. The CF and NEBF correspond to the seismogenic structures activated with the main shock (0 s) and 40 s shock of the 1980 Irpinia earthquake [1], respectively. Furthermore, the central part of the sector bounded by the two structures described above, which has been identified as a major graben structure based on both seismological and levelling data [1,96], is also characterized by similar, slightly higher positive velocity.

In addition, two elongated bands, NW-SE-oriented and characterized by slight subsidence (0 to −1 mm/y or −1 to −2 mm/y mean velocity classes), are identified in the surface projection of hanging wall blocks of the CF and NEBF. These bands form narrow (circa 4–5 km wide) belts following the surface projection of the CF and NEBF major structures.

In the maps derived by ENVISAT PS datasets, the area of interest is, for the most part, subject to subsidence (the most represented is the 0 to −1 mm/y mean velocity class), but areas that in the ERS-based maps show uplift are still visible as slightly uplifting.

In the maps derived from the 'native' datasets (Figures 8 and 9), localized orange or blue spots that are widespread in almost the entire analyzed area can be noticed. Comparison of spatial distribution of orange/blue spots with surface geology features shown in the schematic map of Figure 2a suggests that the spots are most probably related to gravitational phenomena, which affect the shale/clay-dominated formations (e.g., internal units and Pliocene wedge-top basin deposits) that have been cropped out in most of the investigated region.

The tectonic setting of the 1980 Irpinia earthquake region is shown in Figure 2. As already mentioned previously, the Irpinia earthquake was characterized by a complex source mechanism, associated with at least three normal faulting ruptures on distinct fault segments [1] involving the activation of both the Central Fault and the NE Boundary Fault shown in Figure 2b.

Like Peltzer et al. [115], we analyzed intermediate- and near-field postseismic surface displacements following the Irpinia 1980 earthquake using processed ERS 1/2 and ENVISAT SAR data with PS-InSAR technique, covering time intervals between 1992 and 2010. The PSs interpolated maps revealed transient displacement patterns that were either not observed or only partially captured by other geodetic techniques. In particular, the PSs interpolated maps depict vertical displacements of the ground surface. Analysis of the range change maps, which cover an 18-year timespan, started only 12 years after the 1980 earthquake.

Both the ERS and ENVISAT vertical mean velocity deformation maps (Figures 8–12) show that the surface projection of the footwall block to the northeast-dipping CF has uplifted over the analyzed timespan. Such an uplift is more marked during the timespan covered by the ERS satellites (mean velocity of about 1–2 mm/y) than in the 2003–2010 timespan (Figures 12 and 13) On the other hand, the surface projection of the hanging wall block of the CF, near field of the fault itself, is characterized by slow subsidence (in the −1 to 0 mm/y mean velocity range; Figure 12). Likewise, the surface projection of the footwall block of southwest-dipping NEBF has uplifted, more markedly (mean velocity of about 1–2 mm/y) during the timespan covered by the ERS surveys, while the hanging wall block is characterized by slow subsidence (Figures 12 and 13). Uplift, more evident in the timespan covered by the ERS surveys, characterizes the sector located at the northwestern termination of the surface projection of the CF (Figure 12).

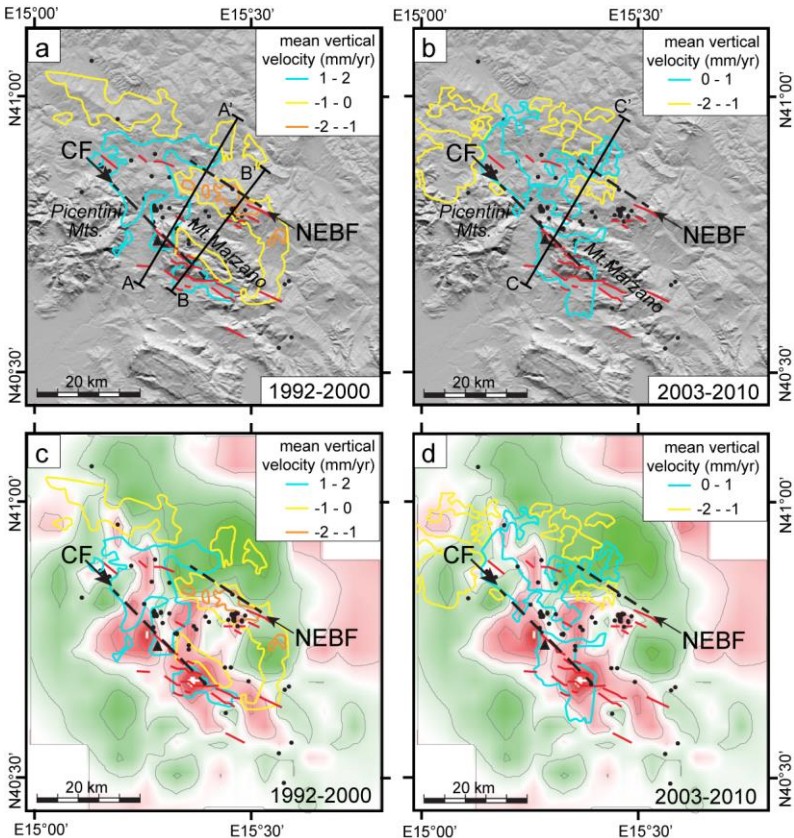

**Figure 12.** Multitemporal ground deformation patterns plotted against the pattern of active faults at the surface (red segments, from Ascione et al. [7]) and the projections at the surface of the deep-seated CF and NEBF (dashed black lines, from Amoroso et al. [8]; see Figure 2), which correspond to the structures activated with the 1980 Irpinia earthquake at 0 s and 40 s, respectively. The ground deformation patterns result from analyses of the ERS and ENVISAT datasets selected by the *Cluster and Outlier Analysis*. (**a**,**c**) Ground deformation in the 1992–2000 timespan. (**b**,**d**) Ground deformation in the 2003–2010 timespan. The map in the background of diagrams (**c**,**d**) corresponds to the spatial distribution of the $V_P/V_S$ ratio in a horizontal slice at 6 km depth through the P-wave tomographic model, with micro-earthquakes (black dots) recorded from 2005 to 2011, as in Figure 7 (from Amoroso et al. [8], modified). Traces of ground deformation profiles of Figure 13 are also shown in diagrams (**a**,**b**).

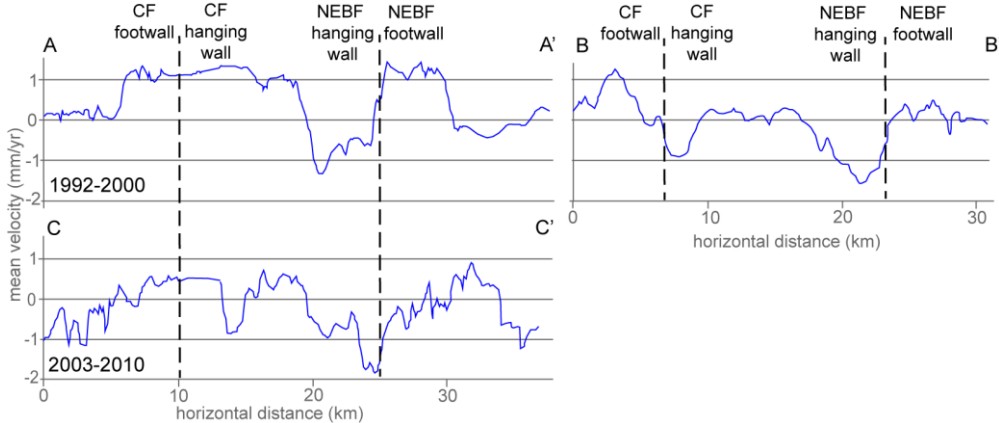

**Figure 13.** Postseismic ground deformation profiles across the 1980 Irpinia earthquake epicentral area (see Figure 12a for locations of profiles A–A' and B–B' and Figure 12b for location of profile C–C'). The dashed black lines indicate the locations of the projections at the surface of the deep-seated CF and NEBF, which correspond to the structures activated with the 1980 Irpinia earthquake at 0 s and 40 s, respectively.

Subsidence of the surface projection of the hanging wall to the NEBF, more pronounced in the 1992–2000 timespan along the southeastern profile (Figure 13), suggests that postseismic afterslip occurred along this major seismogenic fault segment at depth. Surface deformation around the CF shows a more articulated pattern. The southeastern portion of the surface projection of this fault is characterized by a pattern consistent with postseismic afterslip (footwall uplift and hanging wall subsidence; Figure 12a,b; Figure 13). However, to the NW, uplift also characterizes the surface projection of the hanging wall block of the CF. Such a behavior has no straightforward interpretation. As high VP/VS values occur at depth in the region of 'anomalous' uplift in the CF hanging wall (Figure 12c,d), a correlation of subdued, positive ground deformation with fluid accumulation at depth may be hypothesized.

## 9. Concluding Remarks

The analysis of PS-InSAR data from the region struck by the strongest Italian earthquake of the last century, the MS 6.9 1980 Irpinia earthquake, has shown that ground deformation has affected this area in the last decades. The PS-InSAR data analysis covers a timespan ranging from 12 to 30 years after the earthquake. The analysis showed that cumulative deformation is consistent with coseismic deformation inferred from both seismological data (rupture mechanisms of the three main shocks which occurred in a 40 s timespan) [1], levelling data [1,91] and coseismic surface faulting [10,12,18,19]. In addition, it is consistent with evidence of Late Quaternary active faults at the surface (e.g., [7,22]). In particular, evidence for continuing uplift of the footwall—and subsidence of the hanging wall blocks—of the two major faults activated with the 23th November earthquakes has been identified.

The results of PS-InSAR data show that postseismic deformation was still occurring 30 years after the earthquake. The slight decrease in the uplift rate from the timespan surveyed by the ERS satellite to that covered by the ENVISAT could represent the effect of decay in the time fault of creep/low energy earthquakes over decades after the earthquake and of the consequent decrease in postseismic deformation. Within this framework, we considered the ERS and ENVISAT datasets, both of which are high quality and perfectly consistent in terms of long-term trends. Therefore, the observed decay of postseismic deformation was interpreted as real and not induced by the different satellites. In addition, the PS-InSAR data analysis also showed that the region in the mid part between the two main structures activated with the 1980 earthquakes is currently affected by slow uplift. Based on spatial superposition of this region with the rock volume (extending from ~4 km depth downward) that has been identified as saturated of fluids, such uplift may be interpreted as the response, at the surface, to the accumulation of fluid (particularly $CO_2$, based on the widespread evidence discussed in this review paper) at depth. Comparison between the ERS- and ENVISAT-based maps showed that uplift has decreased in the analyzed 20-year time window. However, it is not possible to assess whether such a decrease represents effective diminishing uplift—and possibly decreasing fluid input at depth—or only a stage in a fluctuating long-term process. Previous studies on postseismic deformation elsewhere in the world have suggested that a progressive decay of surface deformation over a 20- to 30-year timespan is compatible with the postseismic afterslip process (e.g., [104]). On the other hand, recent results on high-resolution analysis of microseismicity suggest a multiyear cycle behavior related to $CO_2$-brine accumulation and sealing within a reservoir at hypocentral depths beneath the Irpinia region [116]. According to these studies, when the pressure in the deep $CO_2$-brine reservoir increases, the seismicity tends to progressively be distributed over fault zones. The process culminates with Mw ≥ 3.5 earthquakes, which appears to lead to the creation efficient fracture porosity and related permeability, allowing the fluids to escape and migrate toward the surface. The sequences are then followed by a slow decrease in the stress level until a new stress loading cycle starts. These cycles of stress loading related to fluid accumulation in the subsurface [116] clearly point out that there may be further mechanisms besides postseismic afterslip which is able to account for at least some of the ground motions detected by our InSAR analysis.

The 1980 Irpinia earthquake provides a fundamental lesson on earthquake faulting in the Apennines. Fault segmentation occurs at various scales starting from the seismogenic source. The occurrence of at least three major subevents at 0 s, 20 s, and 40 s implies the activation of multiple deep portions/segments of the crustal-scale seismogenic structure. The strongly anisotropic structure of the allochthonous cover, including various thrust sheets overlying a shale-dominated and fluid-saturated mélange zone resting on top of a more rigid substratum represented by the Apulian Platform, produced an even more complex network of shallow, relatively small (<10 km long) fault strands. This, in turn, resulted in a rather discontinuous and apparently chaotic pattern of surface ruptures, as discussed in this review paper. Within this framework, results from paleoseismological studies obtained from a single fault strand should be used, taking into account that earthquake faulting in the Apennines is clearly a three-dimensional process involving crustal volumes rather than just two-dimensional fault planes. As earthquake-related slip is variably and complexly distributed among various surface fault strands, recurrence intervals and slip rates obtained from a particular fault segment may not be representative of the activity of the seismogenic structure controlling seismicity and the characteristic earthquake for that region. This makes the 'hunting' for the faults that produced historical earthquakes particularly challenging.

**Supplementary Materials:** The following are available online at http://www.mdpi.com/2076-3263/10/12/493/s1, Figure S1: Maps showing the original PS datasets, Figure S2: Spatial selection of PS data, Figure S3: LoS-oriented ascending and descending mean velocity maps in the 1992–2000 time span, Figure S4: LoS-oriented ascending and descending mean velocity maps in the 2003–2010 time span, Figure S5: Comparison between 'PS normal subsets' and 'native' datasets, Figure S6: The three ERS '*marzano'* subsets, Figure S7: The four ENVISAT "Marzano" subsets, Figure S8: Comparison of the results obtained applying the Cluster and Outlier Analysis vs. the Hot Spot Analysis to the ERS PS data subsets, Figure S9: Comparison of the results obtained applying the Cluster and Outlier Analysis vs. the Hot Spot Analysis to the ENVISAT PS data subsets, Figure S10: ERS_T494_F2781_CL001_POTENZA_NORM _D_marzano_CO_2k_IDW and mean velocity map of PSs classified as "not significant", Figure S11: PST2009_ENVISAT_T265_F2781_CL001_AVELLINO_DESCE_NORM_marzano_CO_2k_IDW_nsig and mean velocity map of the PSs classified as "not significant". Table S1: Comparison between the number of PS of the original datasets and those of the subsets obtained through the outliers exclusion process to perform statistical normal distributions of PSs, Table S2: Number of ERS PSs in the subsets deriving from the selection with Cluster and Outlier Analysis mapping tool, Table S3: Number of ENVISAT PSs in the subsets deriving from the selection with Cluster and Outlier Analysis, Table S4, ERS PSs selection with Cluster and Outlier Analysis, Table S5: ENVISAT PSs selection with Cluster and Outlier Analysis.

**Author Contributions:** Conceptualization, A.A. and S.M.; methodology, A.A. and S.N.; software, S.N.; validation, A.A., S.N. and S.M.; formal analysis, S.N.; investigation, A.A., S.N. and S.M.; resources, A.A., S.N. and S.M.; data curation, A.A. and S.N.; writing—original draft preparation, A.A., S.N. and S.M.; writing—review and editing, A.A. and S.M.; visualization, A.A. and S.N.; supervision, A.A. and S.M.; project administration, A.A. and S.M.; funding acquisition, A.A. and S.M. All authors have read and agreed to the published version of the manuscript.

**Funding:** This research received no external funding.

**Acknowledgments:** We wish to thank Aldo Zollo, Gaetano Festa, Matteo Picozzi and Antonio Emolo for fruitful discussions. Thoughtful and constructive reviews by two Anonymous Reviewers are gratefully acknowledged.

**Conflicts of Interest:** The authors declare no conflict of interest.

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
