# Peer review of "The MS 6.9, 1980 Irpinia Earthquake from the Basement to the Surface: A Review of Tectonic Geomorphology and Geophysical Constraints, and New Data on Postseismic Deformation"

_geosciences, doi:10.3390/geosciences10120493_

Round 1
Reviewer 1 Report
The manuscript represents a comprehensive review of the tectonic, seismological and geomorphological framework of area of Southern Apennines hosting the fault system responsible of large earthquakes, including the Ms 6.9 Irpinia earthquake. The work is also integrated with the analysis of new PS-InSAR data covering a time span that ranges from 12 to 30 years after the earthquake. The manuscript is well written and presented; the English quality is good, too. There are only few typos mistakes and minor revisions that can be easily addressed following my comments given below.
Comments:
- Page 5, line 181: Please, add a dot before “In particular”.
- Page 5, line 185: Here I suggest using lowercase for the word “boundary” because it can be considered as the adjective of the plural noun “faults”.
- Page 5, lines 189-193: Reference number 49 was a poster presented in 2010 and not in 2017. Anyway, microearthquake sequences and repeated earthquakes are discussed in detail in their paper published in 2012 (Scientific Reports, doi: 10.1038/srep00410) which was not cited in your review.
- Page 11, line 410: Reference number 92 is Amato and Selvaggi (1993) and not Festa et al. which is not listed in the References. Please, correct the mistake.
- Page 12, line 439; page 13, line 472; page 14, line 499: Please, substitute “c.” with the full word.
- Page 12: The sentence between lines 452-454 is a repetition of the sentence between lines 442-443. Please, rearrange the text and remove one of the two sentences.
- Subsection 7.1 in my opinion is too long and is presented in a slightly different style with respect to other sections. It is not mandatory, but I strongly recommend to move some of the figures and text in the supplements.
- Figures 11-19: Green lines from Ascione et al., 2013 are not visible. Please, use a different color (red, fuchsia?) or a different line representation (dashed line?).
- Figure 19: I suggest to use letters A-D for the four panels. In this way you can avoid the use of words “upper left corner”, “upper right corner”, “lower left corner”, and “lower right corner” both in the figure caption and in the text.
- Page 23, line 728: Please, substitute “con” with “can”.
- Page 23, line 746: Substitute “map” with “maps”.
- Figure 20: Why the two panels on the right are displayed with transparency? Please, standardize them to the two panels on the left. Furthermore, I suggest to list the four panels from the left to the right with letters A-D.
- Page 25: Please, fill “Author Contributions” section and “Acknowledgements” (if any).
Author Response
Camerino, December 3rd, 2020
Manuscript ID: geosciences-1002524
Type of manuscript: Article
Title: The MS 6.9, 1980 Irpinia earthquake from the basement to the surface:
a review of tectonic geomorphology and geophysical constraints, and new data
on postseismic afterslip
Authors: Alessandra Ascione, Sergio Nardò, Stefano Mazzoli *
Dear Editor,
We thank you and the two Referees for the thoughtful and constructive comments, which allowed us to substantially improve our paper.
The manuscript has been revised by taking into account all of the criticisms highlighted by the Reviewers. In the revision notes below, our replies are in blue.
We are confident that this revised version of the paper, improved in line with the Reviewers comments, will fully satisfy the high standards required for publication in Geosciences.
Thank you very much for your kind cooperation.
Best regards,
Stefano Mazzoli
REVIEWER 1
Comments and Suggestions for Authors
The manuscript represents a comprehensive review of the tectonic, seismological and geomorphological framework of area of Southern Apennines hosting the fault system responsible of large earthquakes, including the Ms 6.9 Irpinia earthquake. The work is also integrated with the analysis of new PS-InSAR data covering a time span that ranges from 12 to 30 years after the earthquake. The manuscript is well written and presented; the English quality is good, too. There are only few typos mistakes and minor revisions that can be easily addressed following my comments given below.
We thank Reviewer 1 for the evaluation and appreciation of our work.
Comments:
Page 5, line 181: Please, add a dot before “In particular”.
Done.
Page 5, line 185: Here I suggest using lowercase for the word “boundary” because it can be considered as the adjective of the plural noun “faults”.
Done.
Page 5, lines 189-193: Reference number 49 was a poster presented in 2010 and not in 2017. Anyway, microearthquake sequences and repeated earthquakes are discussed in detail in their paper published in 2012 (Scientific Reports, doi: 10.1038/srep00410) which was not cited in your review.
Done.
Page 11, line 410: Reference number 92 is Amato and Selvaggi (1993) and not Festa et al. which is not listed in the References. Please, correct the mistake.
Done, we fixed this issue (the correct reference was actually another one, which is now included in the revised manuscript).
Page 12, line 439; page 13, line 472; page 14, line 499: Please, substitute “c.” with the full word.
Done.
Page 12: The sentence between lines 452-454 is a repetition of the sentence between lines 442-443. Please, rearrange the text and remove one of the two sentences.
Done.
Subsection 7.1 in my opinion is too long and is presented in a slightly different style with respect to other sections. It is not mandatory, but I strongly recommend to move some of the figures and text in the supplements.
Done. We substantially revised this section and moved several figures and text into the supplementary material.
Figures 11-19: Green lines from Ascione et al., 2013 are not visible. Please, use a different color (red, fuchsia?) or a different line representation (dashed line?).
Done. We corrected these figures.
Figure 19: I suggest to use letters A-D for the four panels. In this way you can avoid the use of words “upper left corner”, “upper right corner”, “lower left corner”, and “lower right corner” both in the figure caption and in the text.
Done.
Page 23, line 728: Please, substitute “con” with “can”.
Done.
Page 23, line 746: Substitute “map” with “maps”.
Done.
Figure 20: Why the two panels on the right are displayed with transparency? Please, standardize them to the two panels on the left. Furthermore, I suggest to list the four panels from the left to the right with letters A-D.
Done.
Page 25: Please, fill “Author Contributions” section and “Acknowledgements” (if any).
Done.

Reviewer 2 Report
General comment
This paper is well written and well organized.
The core of originality is the elaboration of PS from satellite images of ERS and ENVISAT, on a statistical base, in two different nearly contiguous time spans, focused on the Mt. Marzano, and surrounding areas, where the Irpinia M6.9 occurred on 23rd of November 1980.
In my opinion though, the sections preceding the main original (namely section 7) are too densely filled with details. It is a very detailed and well done multidisciplinary overview. It is in general valuable and useful, but most of the details are not strictly needed to let the reader understand your main hypothesis nor to you to support it. I would suggest to “summarize” those sections a little bit.
It is a suggestion that I believe it would enhance the overall readability and let focus on the core of the paper.
Again in terms of length, readability, and details level: the paper is already full of a mix of literature re-elaborated and original images, sometimes from previous papers of the authors themselves. I believe that, to enhance the readability of this valuable work, some of the figures might be at least moved to the auxiliary material. For example all the panoramic pictures might be moved to the supplementary material, especially those not reporting mandatory information or original sketches (eg: Figure 4, Figure 5, Figure 6e,f),
Figures 17 and 18 are repeated in Figure 19 and the size/resolution difference between the two versions does not justify, in my opinion, the duplication. I suggest to keep only the comparative figure just increasing a bit the font size inside the maps.
Comments on the core of the paper
Issue 1: The final interpolated outliers-stripped images present velocities lower than +/-2mm/year in the whole area but in general lower than +/-1 mm/year.
Some spots exceed these values in the “+outliers” images.
Reading the auxiliary material did not clarify to me what final uncertainty you do attribute to the grid points of these interpolated maps.
For what I know there might be uncertainties in such data/elaboration up to 2mm/year especially in low coherent areas and this seem to be in the range of the values reported in your final maps (Figures 17 and 18 and comparative 19 with the exception of spots in the native data maps). I am not saying that the maps that the authors present ARE inside the noise but that it would be important showing to the reader the uncertainty boundaries of the presented results.
In addition: it would be very useful to add one or more vertical SW-NE section(s) across Marzano and Conza, reporting the Z velocity form the interpolated maps of both time-spans (ERS and ENVISAT) possibly with width representing the uncertainty, together with topography profile and faults, above Vp/Vs tomography (like Figure 10b) to let the reader better understanding the spatial-temporal trend across the structure in support of your conclusions.
Issue 2: I think that the hypothesis exposed from line 787 to 791 in the concluding remarks is intriguing. But I’m not sure that you can interpret the differences in the overall amount of vertical displacement in terms of real generalized decrease. You should solve the doubt that this is due to the differences between ERS and ENVISAT. Your potential target readers are not necessarily expert of satellites data and it would be important for them to find such a statement in support of the hypothesis/suggestion, in the main text.
Some minor issues
Line 37: “a combination or both” to “a combination of both”?
Line 91: “(both spatial and in terms of rate)” can be removed. About this: this can be removed like all the not strictly necessary parenthesis. There are in my opinion too many parenthesis in single very long sentences making reading difficult.
Line 94: “shaking and” to “shaking, and” … in general all similar cases should be changed
Line 106: “Such observations form part” to “Such observations are part”
Line 106 to 110: this sentence would be better placed in the Abstract rather than in a middle part of introduction, better clarifying the aim of the paper. Removing it would increase the readability of the introduction.
Line 181: missing “.” after [45,46] I
Line 240: the also?
Line 272: “basins [7,77,78],. “ remove the final comma
Line 305:
- location is in Figure 2c not 3?
- “as it shown by” to “as it is shown by” or “as shown by”
- Anyway please check the whole long sentence from 303 (“On the other hand, […]”) to 307 (“paleoseismological investigations”)
Lines 343 to 347: delete, this is a repeated copy-paste from above.
Line 359: double citation of Figure 5
Line 365: Figure 6 - Trace of this section in map? Report in caption the “resistivity” naming of the “geophysical investigation
Line 406: citation 87 is “Festa, A., Cavagna, S., Barbero, E., Catanzariti, R., & Pini, G. A. Mid-Eocene giant slope failure (sedimentary mélanges) in the Ligurian accretionary wedge (NW Italy) and relationships with tectonics, […]” is this correct?
Line 467 (Figure 8): no legend for the crosses
Line 517 to 522: the sentence from “Surface structures […] to “[…] successions (Figure 2b)” is long and complex to read
Line 532 (Figure 10): dashed lines are grey (as reported in the caption), in Vp/Vs, and white in Vp (as not reported in the caption). Moreover no explanation of 3x3x1 6x6x2 12x12x4 is reported. I know that the explanation is in the Amoroso et al 2014 paper (“Each curve corresponds to a different resolution scale obtained from estimating the resolvability function for each model parame- trization used in the multiscale approach.” but here it is not clear to the reader. Since the caption is exactly the same, though adapted, as the cited paper (including the grey-white issue) I would report also this short explicative sentence in the caption of the present paper.
Line 554: maybe “In this section it is presented” to “In this section we present”
Line 712: Figure is probably 12 not 11
Lines 760 to 764: this single sentence, also including che parenthesis, should be subdivided/simplified.
Line 773: Figure 20, Colors on colors, though nice, is not clear. Transparency unfortunately does not help much. You might use contouring (red and blu?) for up/down, not too densely spaced though, to make this figure clearer.
Lines 821 to 825: You are here stating something important and you are not minimizing anything in my opinion. Only the form is a bit strong and so the following sentence (“By no means we imply that […]”) sounds like an “unrequested apologizing”. I would just rephrase the whole statement in a positive way. It’s your choice anyway. Suggested: “Within this framework, results from paleoseismological studies obtained from a single fault strand should be used taking into account that earthquake faulting in the Apennines is clearly a three-dimensional process involving crustal volumes, rather than two-dimensional fault planes. “
Supplementary
Figure 6a: probably a “there” to remove in the caption.
Figure 7: please switch columns to be coherent with figure 6
Author Response
Camerino, December 3rd, 2020
Manuscript ID: geosciences-1002524
Type of manuscript: Article
Title: The MS 6.9, 1980 Irpinia earthquake from the basement to the surface:
a review of tectonic geomorphology and geophysical constraints, and new data
on postseismic afterslip
Authors: Alessandra Ascione, Sergio Nardò, Stefano Mazzoli *
Dear Editor,
We thank you and the two Referees for the thoughtful and constructive comments, which allowed us to substantially improve our paper.
The manuscript has been revised by taking into account all of the criticisms highlighted by the Reviewers. In the revision notes below, our replies are in blue.
We are confident that this revised version of the paper, improved in line with the Reviewers comments, will fully satisfy the high standards required for publication in Geosciences.
Thank you very much for your kind cooperation.
Best regards,
Stefano Mazzoli
REVIEWER 2
General comment
This paper is well written and well organized.
The core of originality is the elaboration of PS from satellite images of ERS and ENVISAT, on a statistical base, in two different nearly contiguous time spans, focused on the Mt. Marzano, and surrounding areas, where the Irpinia M6.9 occurred on 23rd of November 1980.
We thank Reviewer 1 for the evaluation and appreciation of our work.
In my opinion though, the sections preceding the main original (namely section 7) are too densely filled with details. It is a very detailed and well done multidisciplinary overview. It is in general valuable and useful, but most of the details are not strictly needed to let the reader understand your main hypothesis nor to you to support it. I would suggest to “summarize” those sections a little bit.
It is a suggestion that I believe it would enhance the overall readability and let focus on the core of the paper.
Done. We trimmed those sections,
Again in terms of length, readability, and details level: the paper is already full of a mix of literature re-elaborated and original images, sometimes from previous papers of the authors themselves. I believe that, to enhance the readability of this valuable work, some of the figures might be at least moved to the auxiliary material. For example all the panoramic pictures might be moved to the supplementary material, especially those not reporting mandatory information or original sketches (eg: Figure 4, Figure 5, Figure 6e,f),
Done. We moved those figures into the supplementary material.
Figures 17 and 18 are repeated in Figure 19 and the size/resolution difference between the two versions does not justify, in my opinion, the duplication. I suggest to keep only the comparative figure just increasing a bit the font size inside the maps.
Fixed. We removed those figures.
Comments on the core of the paper
Issue 1: The final interpolated outliers-stripped images present velocities lower than +/-2mm/year in the whole area but in general lower than +/-1 mm/year.
Some spots exceed these values in the “+outliers” images.
Reading the auxiliary material did not clarify to me what final uncertainty you do attribute to the grid points of these interpolated maps.
For what I know there might be uncertainties in such data/elaboration up to 2mm/year especially in low coherent areas and this seem to be in the range of the values reported in your final maps (Figures 17 and 18 and comparative 19 with the exception of spots in the native data maps). I am not saying that the maps that the authors present ARE inside the noise but that it would be important showing to the reader the uncertainty boundaries of the presented results.
We fully took into account these comments in the revised manuscript, where we specified that: “The PS position values for C-band SAR satellites, such as the ERS and ENVISAT satellites, may be affected by a measurement error of ± 3 mm. However, the ERS and ENVISAT PS data from the MATTM database are high quality. Any single pixel in a radar image is selected only if its temporal and geometrical de-correlation values are extremely low. In fact, the coherence values (ranging from 0 to 1) of the PS of the datasets that cover the investigated area are ≥ 0.6, which means that the PS coherence values are high. In other words, the analysed PS are stable with respect to the radar and, thus, their motions are measured with high precision [FERRETTI ET AL., 2001; COSTANTINI ET AL., 2017]. Consistently, the standard deviations of mean velocity values of single PS, calculated over the time spans of the entire records (i.e., the 1991 – 2000 and 2003 – 2010 time spans, respectively), are ≤ 0.5 mm/yr for about the 90% of both datasets”.
In addition: it would be very useful to add one or more vertical SW-NE section(s) across Marzano and Conza, reporting the Z velocity form the interpolated maps of both time-spans (ERS and ENVISAT) possibly with width representing the uncertainty, together with topography profile and faults, above Vp/Vs tomography (like Figure 10b) to let the reader better understanding the spatial-temporal trend across the structure in support of your conclusions.
Done. Such maps and profiles are included in new Figures 13 and 14 of the revised manuscript.
Issue 2: I think that the hypothesis exposed from line 787 to 791 in the concluding remarks is intriguing. But I’m not sure that you can interpret the differences in the overall amount of vertical displacement in terms of real generalized decrease. You should solve the doubt that this is due to the differences between ERS and ENVISAT. Your potential target readers are not necessarily expert of satellites data and it would be important for them to find such a statement in support of the hypothesis/suggestion, in the main text.
We included the requested statement in the revised manuscript, where it is now stated that: “Within this framework, we consider the ERS and ENVISAT datasets, both of which are high quality, perfectly consistent in terms of long-term trends. Therefore, the observed decay of postseismic deformation is interpreted as real and not induced by the different satellites”.
Some minor issues
Line 37: “a combination or both” to “a combination of both”?
Fixed.
Line 91: “(both spatial and in terms of rate)” can be removed. About this: this can be removed like all the not strictly necessary parenthesis. There are in my opinion too many parenthesis in single very long sentences making reading difficult.
Done.
Line 94: “shaking and” to “shaking, and” … in general all similar cases should be changed
Fixed.
Line 106: “Such observations form part” to “Such observations are part”
Done.
Line 106 to 110: this sentence would be better placed in the Abstract rather than in a middle part of introduction, better clarifying the aim of the paper. Removing it would increase the readability of the introduction.
Done.
Line 181: missing “.” after [45,46] I
Fixed.
Line 240: the also?
Fixed.
Line 272: “basins [7,77,78],. “ remove the final comma
Fixed.
Line 305:
location is in Figure 2c not 3?
Fixed.
“as it shown by” to “as it is shown by” or “as shown by”
Fixed.
Anyway please check the whole long sentence from 303 (“On the other hand, […]”) to 307 (“paleoseismological investigations”)
Fixed.
Lines 343 to 347: delete, this is a repeated copy-paste from above.
Fixed.
Line 359: double citation of Figure 5
Fixed.
Line 365: Figure 6 - Trace of this section in map? Report in caption the “resistivity” naming of the “geophysical investigation
Fixed.
Line 406: citation 87 is “Festa, A., Cavagna, S., Barbero, E., Catanzariti, R., & Pini, G. A. Mid-Eocene giant slope failure (sedimentary mélanges) in the Ligurian accretionary wedge (NW Italy) and relationships with tectonics, […]” is this correct?
It was not that, but we fixed it.
Line 467 (Figure 8): no legend for the crosses
Fixed.
Line 517 to 522: the sentence from “Surface structures […] to “[…] successions (Figure 2b)” is long and complex to read
Fixed.
Line 532 (Figure 10): dashed lines are grey (as reported in the caption), in Vp/Vs, and white in Vp (as not reported in the caption). Moreover no explanation of 3x3x1 6x6x2 12x12x4 is reported. I know that the explanation is in the Amoroso et al 2014 paper (“Each curve corresponds to a different resolution scale obtained from estimating the resolvability function for each model parametrization used in the multiscale approach.” but here it is not clear to the reader. Since the caption is exactly the same, though adapted, as the cited paper (including the grey-white issue) I would report also this short explicative sentence in the caption of the present paper.
Done.
Line 554: maybe “In this section it is presented” to “In this section we present”
Fixed.
Line 712: Figure is probably 12 not 11
Fixed.
Lines 760 to 764: this single sentence, also including che parenthesis, should be subdivided/simplified.
Fixed.
Line 773: Figure 20, Colors on colors, though nice, is not clear. Transparency unfortunately does not help much. You might use contouring (red and blu?) for up/down, not too densely spaced though, to make this figure clearer.
Fixed.
Lines 821 to 825: You are here stating something important and you are not minimizing anything in my opinion. Only the form is a bit strong and so the following sentence (“By no means we imply that […]”) sounds like an “unrequested apologizing”. I would just rephrase the whole statement in a positive way. It’s your choice anyway. Suggested: “Within this framework, results from paleoseismological studies obtained from a single fault strand should be used taking into account that earthquake faulting in the Apennines is clearly a three-dimensional process involving crustal volumes, rather than two-dimensional fault planes. “
Fixed, corrected as suggested by the Reviewer.
Supplementary
Figure 6a: probably a “there” to remove in the caption.
Fixed.
Figure 7: please switch columns to be coherent with figure 6
Done.
